# Neck linker docking is critical for Kinesin-1 force generation in cells but at a cost to motor speed and processivity

Breane G Budaitis[1], Shashank Jariwala[2†], Dana N Reinemann[3†], Kristin I Schimert[4†], Guido Scarabelli[2], Barry J Grant[5], David Sept[4,6,7], Matthew J Lang[3,8], Kristen J Verhey[1,4,9]*

[1]Cellular and Molecular Biology Program, University of Michigan, Ann Arbor, United States; [2]Department of Computational Medicine and Bioinformatics, University of Michigan, Ann Arbor, United States; [3]Department of Chemical and Biomolecular Engineering, Vanderbilt University, Nashville, United States; [4]Biophysics Program, University of Michigan, Ann Arbor, United States; [5]Division of Biological Sciences, Section of Molecular Biology, University of California, San Diego, San Diego, United States; [6]Department of Biomedical Engineering, University of Michigan, Ann Arbor, United States; [7]Center for Computational Medicine and Bioinformatics, University of Michigan, Ann Arbor, United States; [8]Department of Molecular Physiology and Biophysics, Vanderbilt University School of Medicine, Nashville, United States; [9]Department of Cell and Developmental Biology, University of Michigan Medical School, Ann Arbor, United States

*For correspondence:
kjverhey@umich.edu

†These authors contributed equally to this work

Competing interests: The authors declare that no competing interests exist.

**Abstract** Kinesin force generation involves ATP-induced docking of the neck linker (NL) along the motor core. However, the roles of the proposed steps of NL docking, cover-neck bundle (CNB) and asparagine latch (N-latch) formation, during force generation are unclear. Furthermore, the necessity of NL docking for transport of membrane-bound cargo in cells has not been tested. We generated kinesin-1 motors impaired in CNB and/or N-latch formation based on molecular dynamics simulations. The mutant motors displayed reduced force output and inability to stall in optical trap assays but exhibited increased speeds, run lengths, and landing rates under unloaded conditions. NL docking thus enhances force production but at a cost to speed and processivity. In cells, teams of mutant motors were hindered in their ability to drive transport of Golgi elements (high-load cargo) but not peroxisomes (low-load cargo). These results demonstrate that the NL serves as a mechanical element for kinesin-1 transport under physiological conditions.
DOI: https://doi.org/10.7554/eLife.44146.001

## Introduction

Kinesin motor proteins are responsible for orchestrating fundamental microtubule-based processes including cell division, intracellular trafficking, cytoskeletal organization, and cilia function (*Hirokawa et al., 2009*; *Verhey and Hammond, 2009*). All kinesins contain a highly-conserved motor domain with signature sequences for nucleotide and microtubule binding. How nucleotide-dependent conformational changes in the catalytic site result in a mechanical output that drives cargo transport has been a fundamental question in the field.

The two motor domains in most dimeric kinesin motors are linked via a flexible 12–18 amino acid sequence called the neck linker (NL) (*Hariharan and Hancock, 2009*; *Kozielski et al., 1997*). The NL has been suggested to serve as a structural element critical for both directed motility and force

generation of kinesin motors. For kinesin-1, the founding member of the kinesin superfamily, structural and spectroscopic studies have shown that conformational changes in the NL are coupled to the nucleotide state of the motor domain. Specifically, the NL undergoes a transformation from being flexible in both the ADP-bound and nucleotide-free states to being docked along the core motor domain in the ATP-bound state (*Rice et al., 1999*; *Rosenfeld et al., 2001*; *Sindelar et al., 2002*; *Skiniotis et al., 2003*; *Asenjo et al., 2006*; *Sindelar and Downing, 2010*; *Gigant et al., 2013*; *Shang et al., 2014*). NL docking of the leading motor domain positions the lagging motor domain forward along the microtubule track, thereby specifying the direction of motility. NL docking also coordinates the alternating ATPase cycles of the two motor domains to ensure processive stepping (*Case et al., 2000*; *Tomishige and Vale, 2000*; *Hahlen et al., 2006*; *Yildiz et al., 2008*; *Clancy et al., 2011*; *Dogan et al., 2015*; *Isojima et al., 2016*; *Liu et al., 2017*). Recent work has extended the model that nucleotide-dependent conformational changes in the NL drive processive stepping to other members of the kinesin superfamily (*Nitta et al., 2008*; *Muthukrishnan et al., 2009*; *Shastry and Hancock, 2010*; *Shastry and Hancock, 2011*; *Atherton et al., 2014*; *Cao et al., 2014*; *Atherton et al., 2017*; *Ren et al., 2018*).

The role of the NL in force generation has been more difficult to discern (*Block, 2007*). ATP-induced NL docking involves distinct interactions of the two β-strands that comprise the NL, β9 and β10 (*Figure 1C,D*). The first half of the NL, β9, pairs with another β-strand, the coverstrand (CS or β0), located at the opposite end of the core motor domain. The zippering of β9 of the NL with β0 of the CS forms a 2-stranded β-sheet, termed the cover-neck bundle (CNB), to provide the power-stroke for force generation by kinesin-1 (*Hwang et al., 2008*; *Khalil et al., 2008*). Support for the CNB as a mechanical element comes from optical trap assays where point mutations in the CS designed to hinder β-strand formation, or deletion of the entire CS, in the fly kinesin-1 motor significantly reduced the motor's ability to withstand load (*Khalil et al., 2008*). CNB formation may be a critical element for force generation across the kinesin superfamily as recent work has shown that the coverstrand (CS or β0) and NL (β9) of members of the kinesin-5 and kinesin-6 families engage in CNB formation in response to ATP binding (*Atherton et al., 2017*; *Hesse et al., 2013*).

After CNB formation, the C-terminal segment of the NL (β10) is predicted to dock along the surface of the core motor domain. In particular, an asparagine residue between β9 and β10 begins the process of docking β10 of the NL onto β7 of the motor core. This asparagine residue (N334) is predicted to serve as a latch (the N-latch) to hold the docked NL along the core motor domain (*Hwang et al., 2008*) but its role in force generation has not been directly tested. The N-latch residue is conserved in most kinesins, particularly motors known to processively step along microtubules (*Figure 1—figure supplement 2*), suggesting that N-latch formation may also be a conserved feature of kinesin force generation.

Whether CNB and/or N-latch formation are critical for multiple kinesin motors to drive transport of membrane-bound cargo under physiological conditions is not known. To address this, we combined molecular dynamics simulations, in vitro single-molecule assays, and cell-based transport assays to delineate how NL docking influences kinesin-1 motors cooperating in teams to transport membrane-bound cargoes in cells. We found that mutations that disrupt CNB formation and/or N-latch formation severely reduced the ability of single kinesin-1 motors to successfully transport against load in an optical trap. Strikingly, single mutant motors traveled faster and for longer distances under unloaded conditions as compared to wild type (WT) motors. These results indicate that mutations to the CS and N-latch of kinesin-1 can enhance processivity and velocity but at a cost to force production. Mutant motors with impaired CNB formation or N-latch formation are able to cooperate to transport low-load cargo in cells. However, the mutant motors are unable to effectively cooperate to transport high-load cargo in cells. Overall, these findings suggest that CNB and N-latch formation are required for transport of high-load cargoes in cells, even when kinesin-1 motors work collectively as a team.

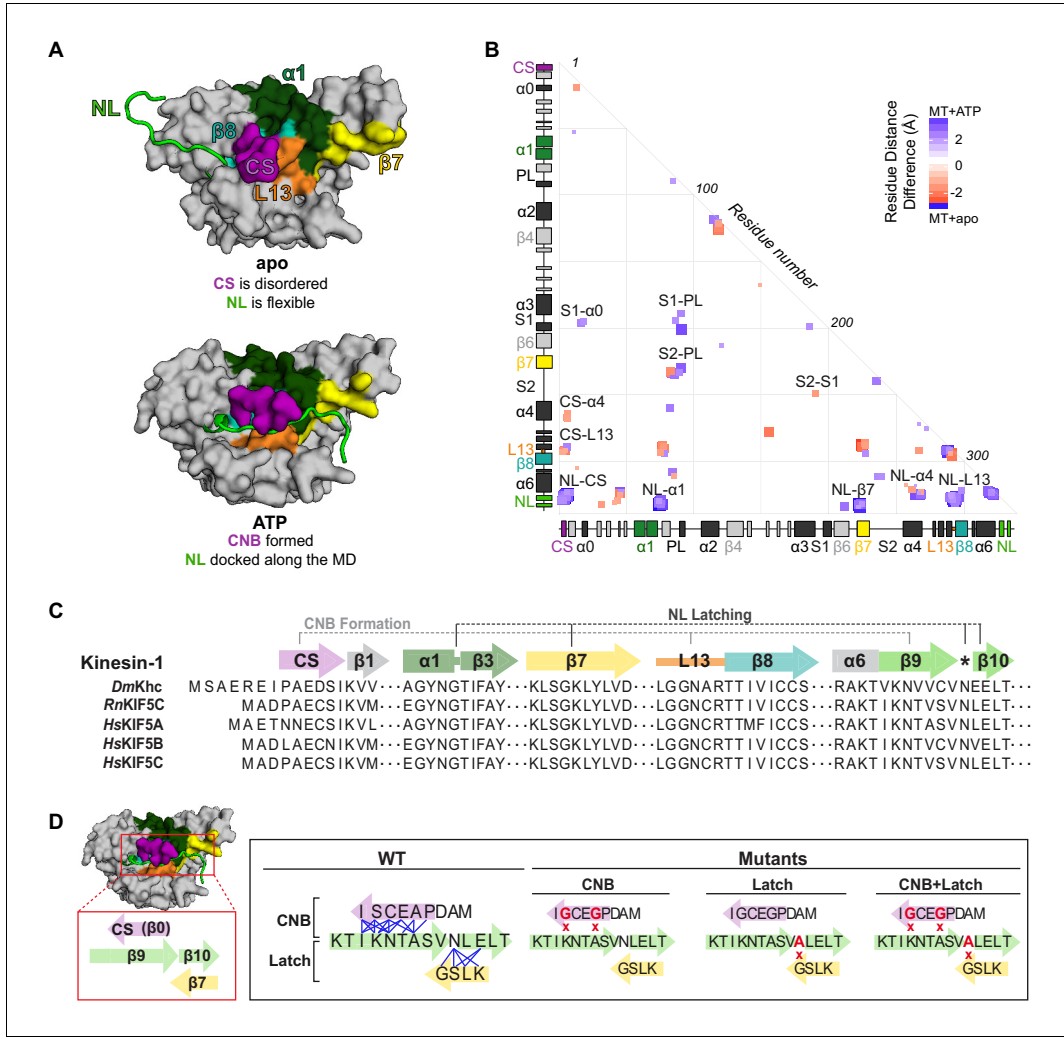

**Figure 1.** MD simulations identify key interactions between the kinesin-1 NL and motor domain. (**A**) Surface representation of the kinesin-1 (*Rn*KIF5C) motor domain in the nucleotide-free (apo) state (top, PDB 4LNU) or ATP-bound, post-power stroke state (bottom, PDB 4HNA). The neck linker (NL, light green) is represented as a cartoon and is flexible in the apo state and docked along the motor domain in the ATP-bound state. Additional secondary structure elements are indicated: coverstrand (CS, purple), α1 (dark green), β7 (yellow), Loop13 (L13, orange), β8 (teal), neck linker (NL: β9 and β10, light green). (**B**) Differences in residue-residue distances between kinesin-1 motors in the apo versus ATP-bound states as determined from MD simulations. The secondary structure elements are laid out along the x- and y-axes with α-helices colored in black, β-strands in grey, or colored according to (**A**). Residue-residue interactions that are significantly closer (p<0.05) in the apo state (red) or ATP-bound state (blue) are indicated on the graph. The magnitude of the distance change is indicated by color intensity. Interactions between key structural elements are labeled. Zoomed views are depicted in *Figure 1—figure supplement 1*. A complete list of residue distance differences are in *Supplementary file 1*. (**C**) Sequence alignment of the kinesin-1 motor domain across species (*Dm*, *Drosophila melanogaster*; *Rn*, *Rattus norvegicus*; *Hs*, *Homo sapiens*). For simplicity, only secondary structure elements indicated in (**A**) are displayed; an asterisk indicates the asparagine-latch (N-Latch). Sequence alignments across kinesin families are in *Figure 1—figure supplement 2*. (**D**) Schematic of key structural elements involved in CNB formation and NL latching in WT and mutant motors. The first-half of the NL (β9, light green) interacts with the C-terminal end of the CS (purple) to form the cover-neck bundle (CNB). The second half of the NL (N-Latch and β10) interacts with β7 (yellow) of the core motor domain for NL docking. Residue-residue contacts for NL docking are depicted as blue lines. Point mutations generated to disrupt CNB formation, N-latch formation, or both are shown in red text.

DOI: https://doi.org/10.7554/eLife.44146.002

The following figure supplements are available for figure 1:

*Figure 1 continued on next page*

*Figure 1 continued*

**Figure supplement 1.** CS and NL interactions in the no nucleotide (apo) and ATP-bound, post-power stroke states of the kineisn-1 motor domain bound to tubulin.
DOI: https://doi.org/10.7554/eLife.44146.003

**Figure supplement 2.** Sequence alignment of the motor domain reveals subtle sequence changes that may alter CNB formation and NL docking across the kinesin superfamily.
DOI: https://doi.org/10.7554/eLife.44146.004

## Results

### Molecular dynamics simulations highlight residues critical for ATP-dependent NL docking

To test whether CNB and/or N-latch formation serve as mechanical elements for kinesin-1 force generation, we sought to identify critical interactions between the CS (β0, aa 4–9), motor core (β1-α6, aa 10–326), and NL (β9-β10, aa 327–338) that we could target for mutagenesis. We performed 100 ns all-atom molecular dynamics (MD) simulations of the rat kinesin-1 (*Rn*KIF5C) motor domain in association with tubulin. Four replicate simulations were carried out for motors in the nucleotide-free (apo) state (PDB 4LNU [*Cao et al., 2014*]) and the ATP-bound state (PDB 4HNA [*Gigant et al., 2013*]), similar to previous analyses of kinesin-5 (*Muretta et al., 2018*). We then compared residue-residue interactions between the apo and ATP-bound states (*Figure 1B*; *Supplementary file 1*) with analysis across replicates to predict statistically significant distance differences (p<0.05). In the apo state (*Cao et al., 2014*), the NL is flexible (*Figure 1A* top) and forms few interactions with the motor domain (*Figure 1B*) while the CS interacts with residues in α4 and in Loop13 (*Figure 1B*, red boxes marked CS-α4 and CS-L13). Specifically, the C-terminal residue (CTR) of the CS (I9) points down into a hydrophobic pocket called the docking pocket (*Sindelar, 2011*) where it contacts residues I266, L269, and A270 of α4 and the remaining residues of the CS contact Loop13 (*Figure 1—figure supplement 1A,B*). Collectively, these interactions sterically block the NL from accessing the docking pocket.

In the ATP-bound state (*Gigant et al., 2013*), the NL is docked along the core motor domain, with each half of the NL (β9 and β10) forming contacts with distinct structural elements (*Figure 1A* bottom, *Figure 1C*, *Video 1*). For the N-terminal half of the NL, β9 forms contacts with the CS to form the cover-neck bundle (CNB) (*Figure 1B*, blue box marked NL-CS) as well as contacts with α4 and Loop13 (*Figure 1B*, blue boxes marked NL-α4 and NL-L13). These contacts are made possible by the ATP-dependent formation of an extra turn at the end of α6, the NL initiation sequence (NIS [*Nitta et al., 2008*]), that positions β9 for insertion between the CS and α4 (*Figure 1—figure supplement 1C,D* [*Sindelar, 2011*; *Lang and Hwang, 2010*]). The first residue of β9 (I327) now occupies the docking pocket and forms contacts with residues I266, L269, and A270 of α4 (*Figure 1—figure supplement 1C,D*). The remaining residues of β9 interact with the CS via a series of backbone-backbone interactions to complete CNB formation (*Figure 1—figure supplement 1C,D*). For the C-terminal half of the NL, β10 docks along

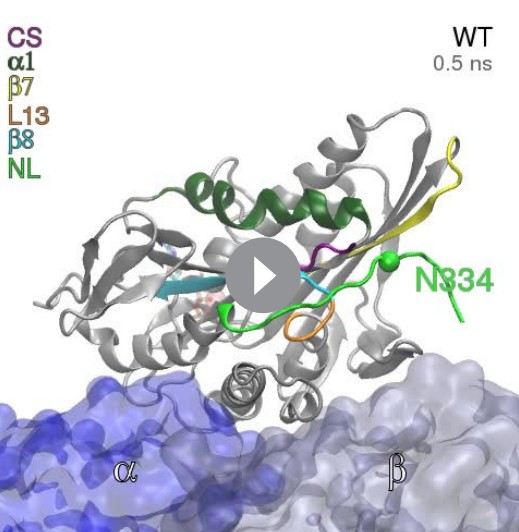

**Video 1.** NL docking in WT kinesin-1. Representative simulation of WT kinesin-1 KIF5C motor domain in the ATP-bound, post-power stroke state. The N-terminal half of the NL (β9) and the N-latch residue N334 (shown as a sphere) are docked along the core motor domain and the C-terminal half of the NL (β10) is dynamic. Secondary structure elements are colored as: coverstrand (CS, purple), α1 (dark green), β7 (yellow), Loop13 (L13, orange), β8 (teal), neck liker (NL: β9 and β10, light green), with α-tubulin (blue) and β-tubulin (light blue).
DOI: https://doi.org/10.7554/eLife.44146.005

the core motor domain through interactions with α1 and β7 (*Figure 1B*, blue boxes marked NL-α1 and NL-β7). Specifically, the N-latch residue (N334) forms interactions with residues E76 and G77 in α1 and residues S225 and L224 in β7 and the remaining residues of β10 provide further backbone interactions with β7 to complete NL docking (*Figure 1—figure supplement 1C,E*).

Overall, the MD simulations build on previous work (*Gigant et al., 2013*; *Nitta et al., 2008*; *Cao et al., 2014*; *Hwang et al., 2008*; *Khalil et al., 2008*; *Hwang et al., 2017*) and identify several residues critical for regulating the flexible-to-docked transformation of the NL. First, the CTR of the CS (I9) occupies the docking pocket bordered by α6, α4, and L13 in the nucleotide-free state such that the NL remains undocked and flexible. Second, I327 of the NIS occupies this pocket in the ATP-bound state and begins the process of NL docking along the core motor domain. We note that the presence of an isoleucine or valine as the CTR and NIS residues is a conserved feature of most kinesin motors that undergo processive motility (*Figure 1—figure supplement 2*). This conservation suggests that the ability of these residues to occupy the docking pocket in a mutually exclusive manner may be a conserved mechanism for kinesin motility and force generation. Third, residue N334 interacts with both α1 and β7 to position the NL along the core motor domain, thereby specifying the direction of motion. As noted previously (*Hwang et al., 2008*; *Khalil et al., 2008*), an asparagine residue between β9 and β10 is a conserved feature of many kinesin motors with an N-terminal motor domain (*Figure 1C* asterisk, *Figure 1—figure supplement 2*).

## CNB and N-latch mutations severely cripple single kinesin-1 motor stepping under load

To delineate the importance of CNB formation and N-latch formation for force generation and transport by kinesin-1 motors, we generated mutations that would weaken the CNB, N-latch, or both. To test the role of CNB formation, CS residues A5 and S8 were mutated to glycine residues (*Figure 1D*, CNB mutant), which have a low propensity to form a β-sheet (*Minor and Kim, 1994*). The A5G/S8G double mutation was previously reported to impair force generation for single *Drosophila melanogaster* kinesin-1 motors in optical trap experiments (*Khalil et al., 2008*). Whether the analogous mutations alter the force generation and/or motility of mammalian kinesin-1 motors has not been tested. To test the role of the N-latch, residue N334 was mutated to an alanine residue (*Figure 1D*, Latch mutant). CNB mutations were also combined with the Latch mutation to assess the importance of CNB formation followed by NL docking in tandem (*Figure 1D*, CNB+Latch mutant).

To verify the effects of the mutations, we carried out MD simulations of the Latch and CNB +Latch mutant motors in the tubulin- and ATP-bound state (post-power stroke) (PDB 4HNA [*Gigant et al., 2013*]). For the Latch mutant, the simulations predict that the N-latch and β10 residues make fewer interactions with α1 and β7 (*Figure 2—figure supplement 1B–D*, *Video 2*). For the CNB+Latch mutant, the simulations predict that mutation of the CS (A5G,S8G) results in intra-CS interactions (*Figure 2D,E*, *Video 3*) rather than interactions with β9 of the NL (*Figure 2A,B*) and that mutation of the N-latch residue (N334A) results in interactions of β10 with the CS and β8 (*Figure 2D,F*, *Video 3*) rather than with α1 and β7 (*Figure 2A,C*). Thus,

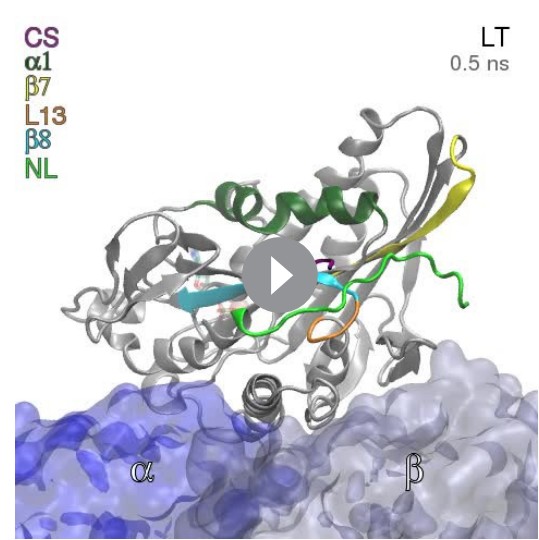

**Video 2.** NL undocking in the Latch mutant. Representative simulation of the Latch mutant motor domain in the ATP-bound state. The C-terminal half of the NL (N-latch residue N334 and β10) undock followed by complete undocking of the NL. Secondary structure elements are colored as: coverstrand (CS, purple), α1 (dark green), β7 (yellow), Loop13 (L13, orange), β8 (teal), neck liker (NL: β9 and β10, light green), with α-tubulin (blue) and β-tubulin (light blue). The changes in residue-residue distances in the Latch mutant compared to the WT are listed in *Supplementary file 2*.

DOI: https://doi.org/10.7554/eLife.44146.008

mutations of CS and N-latch residues weaken CNB formation and NL latching, respectively.

We used a custom-built optical trap apparatus with nanometer-level spatial resolution to assess the effect of the CNB and Latch mutations on kinesin-1's force output. COS7 cell lysates containing FLAG-tagged, constitutively-active versions of WT [$Rn$KIF5C(1-560)] or mutant kinesin-1 motors were subjected to standard single-molecule trapping assays (*Figure 3A*, *Svoboda and Block, 1994*; *Reinemann et al., 2017*; *Reinemann et al., 2018*). Individual WT motors were motile in the absence of load, stalled on the microtubule when approaching the detachment force, and detached from the microtubule at an average force of 4.6 ± 0.8 pN (*Figure 3B,C*), consistent with previous studies (*Khalil et al., 2008*; *Svoboda and Block, 1994*). In contrast, the CNB mutant detached from the microtubule before stalling (*Figure 3C*) and at much lower loads than WT motors (mean detachment force 0.91 ± 0.6 pN, *Figure 3B*), overall similar to the behavior of the fly kinesin-1 with identical mutations in the CS (*Khalil et al., 2008*).

The Latch mutant was also sensitive to small opposing forces exerted by the trap. We found that motors with a weakened N-latch (Latch mutant) did not stall under load (*Figure 3C*) and detached from the microtubule when subjected to a mean force of 0.84 ± 0.4 pN (*Figure 3B*), similar to the CNB mutant motor. This is consistent with previous MD simulations where forced rupturing of the N-Latch led to the rapid unbinding of the entire NL from the core motor domain (*Hwang et al., 2008*). Thus, mutations that weaken either CNB or N-latch formation resulted in motors equally impaired in their ability to drive bead motility under load. The effects of the CNB and Latch mutations were not additive as individual CNB+Latch mutant motors displayed behaviors similar to the CNB and Latch motors: a tendency to detach rather than stall when subjected to load (*Figure 3C*) and detachment from the microtubule at low loads (mean detachment force 0.81 ± 0.5 pN, *Figure 3B*). These results indicate that both CNB formation and N-latch formation are critical for single kinesin-1 motors to generate a strong power stroke and transport continuously under load.

## CNB and latch mutants display enhanced unloaded motility properties

We used single-molecule motility assays to examine the behavior of the CNB and Latch mutants under unloaded conditions. Cell lysates containing kinesin-1 KIF5C(1-560) motors tagged with three tandem monomeric citrine fluorescent proteins (3xmCit) were added to flow chambers containing polymerized microtubules and their single-molecule motility was examined using total internal reflection fluorescence (TIRF) microscopy. The velocity, run length, and microtubule landing rate were determined from kymograph analysis with time displayed horizontally and distance vertically (*Figure 4A*). At least 250 motility events were quantified for each motor across three independent trials and summarized as a histogram or dot plot (*Figure 4B–D*).

Although weakening of the CNB, N-Latch, or both severely diminished the ability of the mutant motors to bear load in the optical trap assay (*Figure 3*), remarkably, all mutant motors were faster and more processive than the WT motor under unloaded conditions. CNB, Latch, and CNB+Latch motors displayed faster velocities of 0.771 ± 0.004 µm/s, 0.761 ± 0.005 µm/s, and 0.788 ± 0.005 µm/s, respectively, compared to 0.617 ± 0.005 µm/s for WT motors (*Figure 4B*). The mutant motors also displayed longer run lengths of 2.07 ± 0.057 µm, 4.27 ± 0.073 µm, and 5.332 ± 0.096 µm, respectively, as compared to 0.990 ± 0.039 µm for WT motors (*Figure 4C*). Examination of the kymographs indicated an increase in the number of motility events for the mutant motors. We therefore quantified how often motors landed on a microtubule to start a processive run (landing rate) and measured landing rates of 0.525 ± 0.01, 1.463 ± 0.03, and 2.442 ± 0.6 events/$\mu m^{-1} nM^{-1} s^{-1}$, respectively, compared with WT motor rate of 0.172 ± 0.006 events/$\mu m^{-1} nM^{-1} s^{-1}$ (*Figure 4D*).

Examination of the kymographs also indicated that the Latch and CNB+Latch mutant motors displayed small gaps between runs (*Figure 4E*). One possibility is that the gaps indicate the reattachment of motors such that multiple runs are joined into superprocessive runs. An alternative possibility is that the mutant motors are superprocessive with the gaps in the runs due to blinking of the fluorescent tag. To distinguish between these possibilities, we compared the single-molecule motility behavior of the CNB+Latch mutant when tagged with a fluorescent marker that does (mRuby) or does not (HaloTag with JF549 dye) exhibit blinking behavior (*Figure 4—figure supplement 1A*). As a control, we carried out the same analysis for a known superprocessive motor, the kinesin-3 motor KIF1A (*Soppina et al., 2014*) (*Figure 4—figure supplement 1A*). For the CNB+Latch mutant motors, the distance moved during a gap was ~1 pixel less than the distance expected for a motor undergoing constant motility (*Figure 4—figure supplement 1D*), whereas for

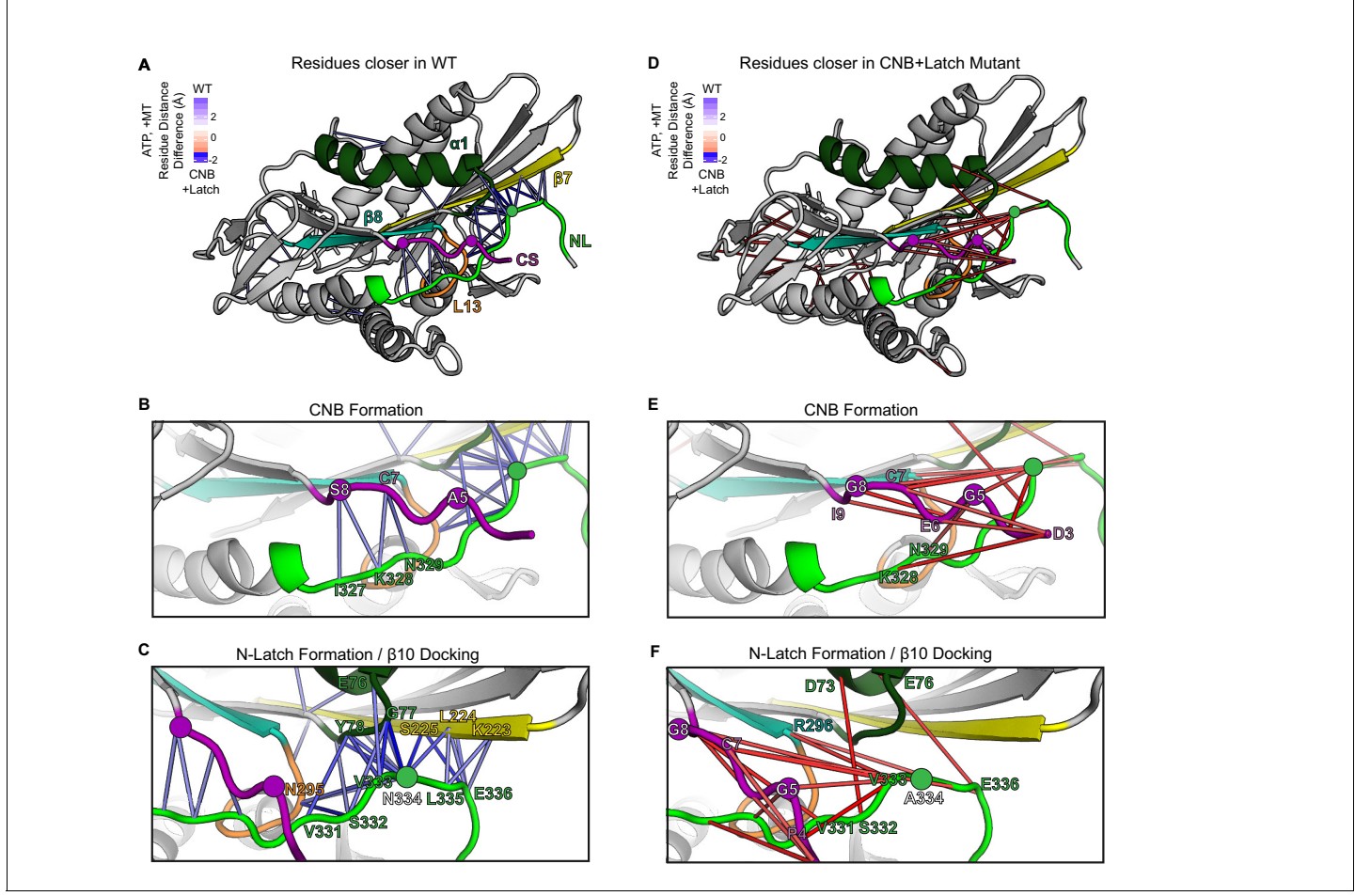

**Figure 2.** MD simulations predict that CNB+Latch mutations alter CNB formation and NL docking. (A–F) The kinesin-1 motor domain in the ATP-bound, post-power stroke state is shown as a cartoon representation (PDB 4HNA). Secondary structure elements are colored: coverstrand (CS, purple), α1 (dark green), β7 (yellow), Loop13 (L13, orange), β8 (teal), neck liker (NL: β9 and β10, light green). Residues targeted for mutations are indicated as circles. (A) Blue lines depict residue-residue that are significantly (p<0.05) closer in the WT motor as compared to the CNB+Latch mutant across replicate MD simulations. The magnitude of the distance change is indicated by color intensity. (D) Red lines depict residue-residue that are significantly (p<0.05) closer in the CNB+Latch mutant as compared to the WT motor across replicate MD simulations. The magnitude of the distance change is indicated by color intensity. A similar comparison between WT and Latch mutant motors is described in *Figure 2—figure supplement 1*. (B, E) Enlarged view of CNB interactions. (B) Contacts between the CS (residues S8, C7) and the NL (β9 residues I327, K328, N329) are shorter in the WT motor, suggesting that CNB formation is disrupted in the CNB+Latch mutant. (E) The mutated CS makes intra-CS contacts rather than interactions with the NL. (C,F) Enlarged view of NL-β7 interactions. (C) The WT motor shows shorter contacts for (i) the N-latch (N334) with β7 (L224, S225) and α1 (G77, Y78) residues, (ii) the N-terminal half of the NL (β9 residues V331, S332, V333) with the core motor domain (L13 residue N295 and α1 residues E76, G77, Y78), and (iii) the C-terminal half of the NL (β10 residue E336) with the core motor domain (β7 residues L224, S225). This suggests that NL docking is disrupted in the CNB+Latch mutant. (F) The mutated NL makes interactions with the CS rather than β7.

DOI: https://doi.org/10.7554/eLife.44146.006

The following figure supplement is available for figure 2:

**Figure supplement 1.** MD simulations predict that mutations of the N-Latch alter CNB formation and NL docking.
DOI: https://doi.org/10.7554/eLife.44146.007

KIF1A motors, the distance moved during a gap was nearly identical to the distance expected for a motor undergoing constant velocity (*Figure 4—figure supplement 1D*). These data are consistent with the idea that the gaps in the kymographs are due to CNB+Latch mutant motors undergoing detachment and reattachment events rather than constant motility. However, we cannot rule out the possibility that the gaps in the kymographs are due to blinking of the fluorescent markers used to track CNB+Latch motors given that i) the distances moved during the gaps are at the limit of resolution of our microscope system and ii) the fluorescent markers may behave differently when attached

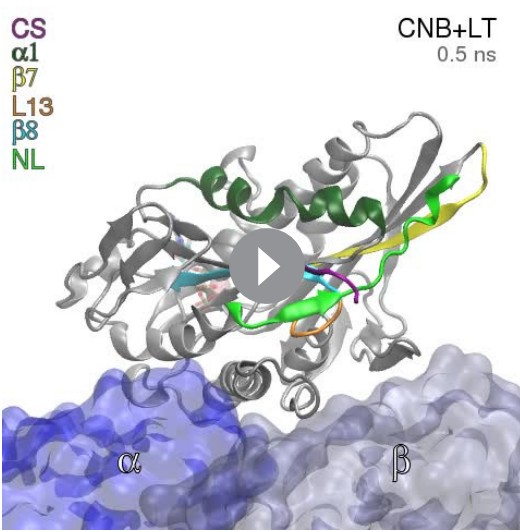

CS
α1
β7
L13
β8
NL

CNB+LT
0.5 ns

α    β

**Video 3.** NL undocking and disruption of CNB in CNB +Latch mutant. Representative simulation of CNB +Latch mutant motor domain in the ATP-bound state. The C-terminal half of the NL (N-latch residue N334 and β10) undocks from the core motor domain followed by disruption of the CNB. Secondary structure elements are colored as: coverstrand (CS, purple), α1 (dark green), β7 (yellow), Loop13 (L13, orange), β8 (teal), neck liker (NL: β9 and β10, light green), with α-tubulin (blue) and β-tubulin (light blue). The changes in residue-residue distances in the CNB+Latch mutant compared to the WT are listed in **Supplementary file 3**.

DOI: https://doi.org/10.7554/eLife.44146.009

to the CNB+Latch mutant motor vs the KIF1A motor. Regardless of whether the gaps in the kymographs are due to reattachment events that string together multiple runs or due to blinking behavior during superprocessive runs, the single-molecule motility data highlight the differences in motor behavior under unloaded and loaded conditions. For kinesin-1 motors, mutations that result in weakened CNB and/or N-latch formation lead to a decreased detachment from the microtubule track (increased run length) under unloaded single-molecule conditions (**Figure 4**) but a more rapid detachment from the microtubule when subjected to a load (**Figure 3**).

## MD simulations predict that modulating CNB and N-latch formation enhances microtubule binding and catalytic site closure

We hypothesized that the enhanced motility properties of the mutant motors under unloaded conditions are due to allosteric effects of mutations designed to hinder NL docking on the nucleotide and microtubule binding regions of the motor domain. We thus re-examined the MD simulations of the Latch and CNB+Latch mutant motors associated with tubulin in the ATP-bound state (PDB 4HNA [**Gigant et al., 2013**]) with a focus on residue interactions outside of the CNB and NL docking regions.

The MD simulations revealed enhanced interactions between elements important for coordinating and hydrolyzing nucleotide in the CNB +Latch mutant as compared to the WT motor (**Figure 5A,B**). Specifically, the residue-residue distances are shorter between the P-loop and α0 (**Figure 5B**, red box PL-α0; **Figure 5—figure supplement 1D–F**; **Supplementary file 3**: E22-S89 distance 4.40 ± 1.29 Å in CNB+Latch versus 7.49 ± 1.97 Å in WT). As the P-loop coordinates ATP in the nucleotide pocket and α0 gates ATP binding (**Hwang et al., 2017**), this result suggests that modulating NL docking influences the ability to capture and/or hold ATP in the nucleotide-binding pocket. Shorter residue-residue distances are also observed between switch one and α0 (**Figure 5B**, red box S1-α0; **Figure 5—figure supplement 1D–F**; **Supplementary file 3**: R25-M198 distance 3.92 ± 0.56 Å in CNB+Latch versus 7.38 ± 2.28 Å in WT) and between switch 1 and switch 2 (**Figure 5B**, red box S1-S2; **Figure 5—figure supplement 1D–F**; **Supplementary file 3**: T196-E237 distance 4.35 ± 0.64 Å in CNB+Latch versus 6.97 ± 1.38 Å in WT). Enhanced interactions between residues involved in coordinating and hydrolyzing nucleotide are also observed in the Latch mutant (**Figure 5—figure supplement 1G–I**; **Supplementary file 2**). As closure of the switch regions is necessary for ATP hydrolysis (**Clancy et al., 2011**; **Cao et al., 2014**; **Turner et al., 2001**; **Parke et al., 2010**), these results indicate that the Latch and CNB+Latch mutations result in enhanced catalytic site closure and ATP hydrolysis that could account for the increase in velocity of the mutant motors under single-molecule, unloaded conditions.

To gain an understanding of how mutations that hinder CNB formation and/or NL docking can result in enhanced microtubule binding (landing rate) and processivity of the mutant motors, we used principle component analysis (PCA) to create a map of the conformational differences of the microtubule-binding surface of the kinesin-1 motor domain in the microtubule-free (and ADP-bound) state as compared to the microtubule-bound (and ATP-bound) state. The structures of seventeen motor domains from five different kinesins (**Supplementary file 4**) were subjected to interconformer analysis with PCA. The CS and NL regions were excluded from the analysis due to their absence

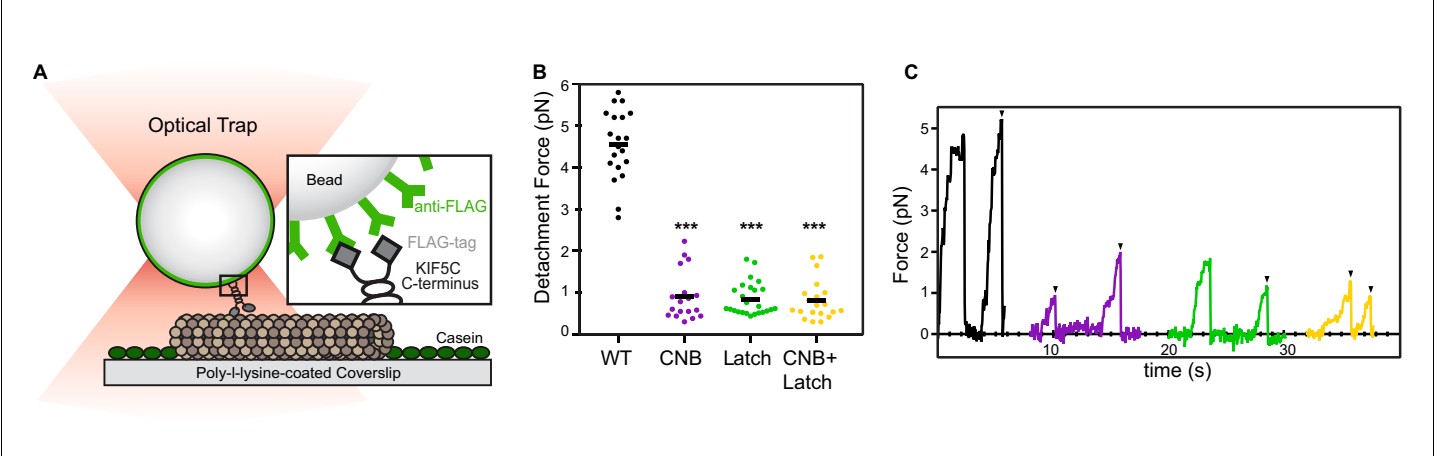

**Figure 3.** CNB and N-Latch formation are critical for force generation by single kinesin-1 motors. (**A**) Schematic of single-molecule optical trap assay. Cell lysates containing FLAG-tagged KIF5C(1-560) motors were incubated with beads functionalized with anti-FLAG antibodies and subjected to standard optical trapping assays. (**B,C**) Force generation of WT (black), CNB (purple), Latch (green), and CNB+Latch (yellow) motors under single-molecule conditions. (**B**) Detachment forces are plotted as a dot plot where each dot indicates the maximum detachment force of an event and the mean for each construct is indicated by a black horizontal line. Maximum detachment forces include motility events where single motors reached a plateau stall before detachment and events where the motor abruptly detached from the microtubule. N ≥ 20 events for each construct; ***, p<0.001, compared to the WT motor. (**C**) Representative traces. Black arrowheads indicate abrupt detachment events.
DOI: https://doi.org/10.7554/eLife.44146.010

from most ADP-bound structures. PCA analysis revealed that the first two dimensions account for over 80% of the variance in atomic positional displacements of the microtubule-binding surface between these states (PC1 79.66%, PC2 4.95%, *Figure 5C*). Thus, PC1 and PC2 provide a suitable conformational space to describe the structural dynamics of kinesin motor domains transitioning from an ADP-bound, microtubule-free state to an ATP-bound, microtubule-bound state. The major conformational difference between these states can be described by PC1 which involves a displacement of α4, where α4 is in a 'down' orientation in the ADP-like, microtubule-free structures and in an 'up' orientation in the ATP-like, microtubule-bound structures (*Figure 5C*), consistent with previous studies (*Scarabelli and Grant, 2013*).

We then used the PCA conformational space to compare how often the WT and CNB+Latch motor domains could adopt the ATP-bound, microtubule-bound state from the ADP-bound, microtubule-free state. MD simulations of WT and CNB+Latch mutant motors were done in replicate for a total of 1 μs each. The conformations sampled by each motor were then projected onto the PCA map. Starting from the PDB 2KIN structure which partially adopts an ATP-like conformation (docked NL but weak microtubule binding [*Sack et al., 1997*]), the WT motor sampled a conformational space between the ADP-bound, microtubule-free and the ATP-bound, microtubule-bound states (*Figure 5C*, blue topographic lines). Starting from the same 2KIN structure, the CNB+Latch mutant sampled an additional conformational space closer to that defined by the ATP-bound, microtubule-bound kinesin structures (*Figure 5C*, red topographic lines). Overall, this suggests that the CNB+Latch mutant has a higher degree of structural flexibility in its microtubule-binding regions as compared to the WT motor domain. This structural flexibility would enable the motor to more readily adopt a conformation compatible with strong microtubule binding in response to ATP in the nucleotide pocket and could account for the enhanced microtubule-landing rate and processivity observed in single-molecule assays.

## CNB and N-latch mutations do not affect the ability of teams of kinesin-1 motors to transport low-load cargo (peroxisomes) in cells

Having defined the force generating properties of individual CNB, Latch, and CNB+Latch mutant motors, we sought to test whether the integrity of CNB formation followed by NL latching is a critical determinant for kinesin motors working in teams to drive cargo transport in cells. To do this, we used an inducible recruitment strategy (*Kapitein et al., 2010*) to link teams of motors to the surface

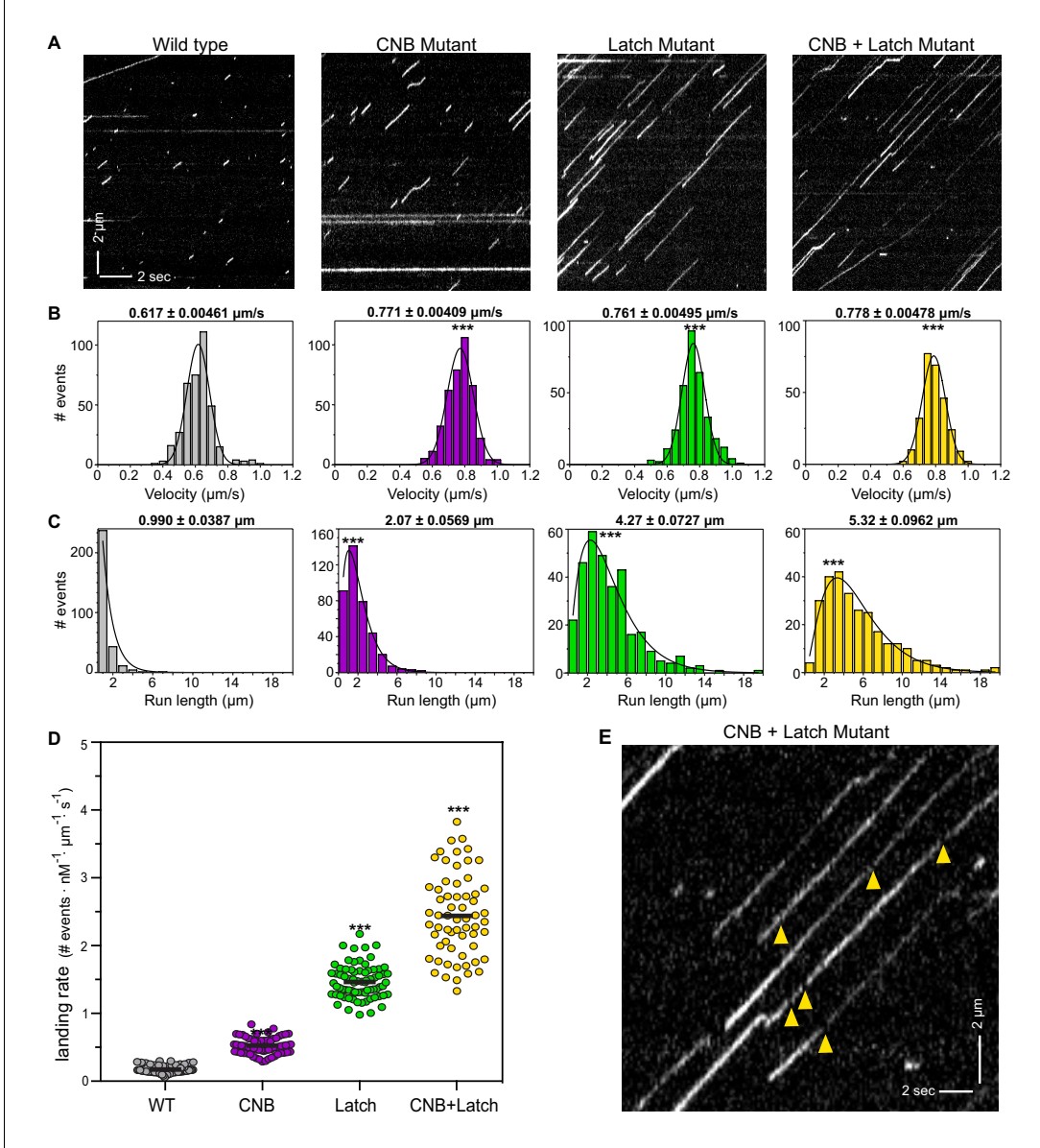

**Figure 4.** CNB and Latch mutants display enhanced motility properties under single-molecule, unloaded conditions. (A) Motility properties of WT or mutant motors tagged with three tandem monomeric Citrines (3xmCit) at their C-termini were analyzed in standard single-molecule motility assays using TIRF microscopy. Representative kymographs are shown with time displayed on the x-axis (bar, 2 s) and distance displayed on the y-axis (bar, 2 µm). (B–D) Quantification of motility properties. From the kymographs, single-motor (B) velocities, (C) run lengths, and (D) landing rates were determined and the data for each population is plotted as a histogram. (B,C) The mean ± SEM are indicated above each graph; N ≥ 250 events across three independent experiments for each construct; ***, p<0.001 as compared to the WT motor. (D) Each dot indicates the landing rate of a single motor with the mean indicated by horizontal black line; N ≥ 250 events across three independent experiments for each construct; ***, p>0.001 as compared to the WT motor. (E) Magnified view of the representative kymograph of the CNB+Latch mutant shown in (A) (y-axis bar, 2 µm; x-axis bar, 2 s); yellow arrowheads indicate gaps in the runs. The gaps are characterized in *Figure 4—figure supplement 1*.
DOI: https://doi.org/10.7554/eLife.44146.011

The following figure supplement is available for figure 4:

**Figure supplement 1.** CNB+Latch mutants exhibit fast reattachment events during processive runs.
DOI: https://doi.org/10.7554/eLife.44146.012

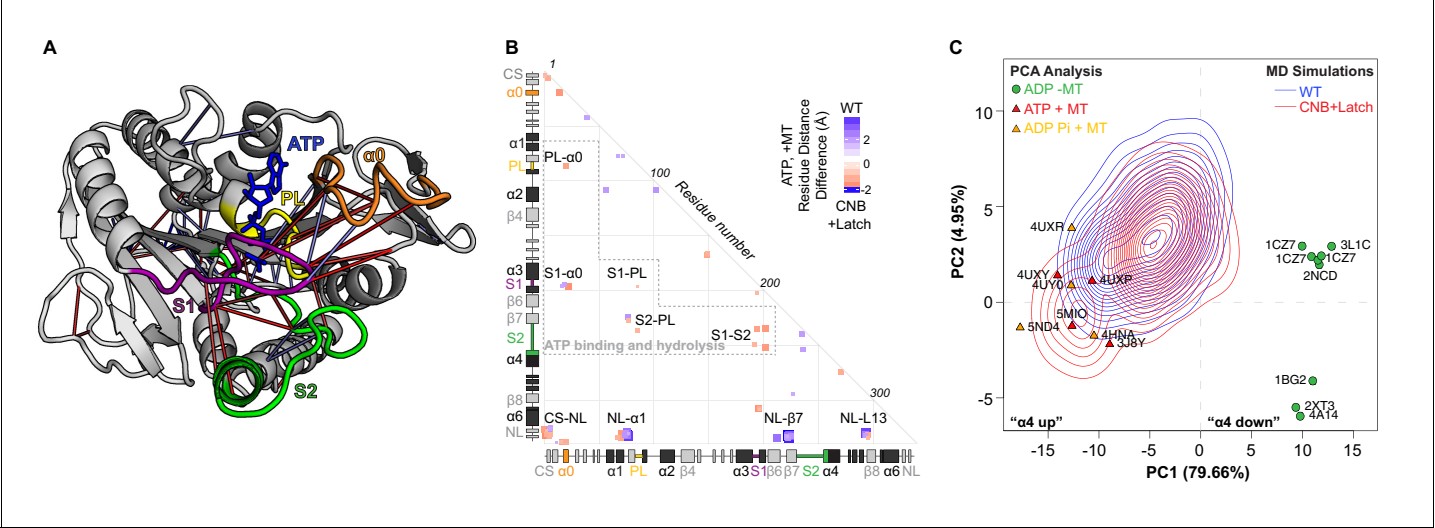

**Figure 5.** CNB+Latch mutations enhance microtubule binding and catalytic site closure. (**A**) Ribbon representation of the kinesin-1 motor domain in the ATP-bound, post-power stroke state (PDB 4HNA). Secondary structure elements critical for nucleotide binding and hydrolysis are colored as follows: Switch 1 (S1, purple), Switch 2 (S2, green), P-loop (yellow), and α0 (orange). Red lines depict residue-residue distances that are shorter in the CNB +Latch mutant motor versus WT motor (p<0.05); blue lines depict residue-residue distances that are shorter in the WT motor versus CNB+Latch mutant motor (p<0.05). The magnitude of the distance change is indicated by line color intensity. Note that the point of view is rotated with respect to previous figures. A more detailed comparison of residue-residue distances within structural elements critical for nucleotide binding and hydrolysis for WT, Latch, and CNB+Latch motors are in *Figure 5—figure supplement 1*. (**B**) Differences in residue-residue distances between WT kinesin-1 and CNB +Latch mutant motors in the ATP-bound, tubulin-bound state in MD simulations. The secondary structure elements are laid out along the x- and y-axes with α- helices in black, β-strands in grey, or colored according to (**A**). Distances that are significantly shorter (p<0.05) in CNB+Latch (red) or WT (blue) motor are displayed. The magnitude of the distance changes is indicated by color intensity; interactions between structural elements are labeled. (**C**) Principle component analysis (PCA) was used to define states of the microtubule-binding surface of kinesin-1. The x-ray structures of seventeen motor domains from five different kinesin families (listed in *Supplementary file 4*) in the ADP-bound or ATP-bound states were utilized. The position of each motor domain structure within the PCA map is indicated together with its nucleotide state (red, ATP; yellow, ADP-Pi; green, ADP), microtubule state (circle, no microtubule; triangle, bound to microtubule), and PDB code. The first two principle components (PC1 and PC2) represent over 80% of the structural variation across the microtubule-binding surface between the ADP-bound and ATP-bound states. PC1 represents the positioning of α4 as 'down' in the ADP-like state and 'up' in the ATP-like state. The ability of WT versus CNB+Latch mutant motors to sample these states was then analyzed by MD simulations starting from the 2KIN structure in the ADP-bound and microtubule-free state. The conformational space explored by each motor is projected as a topographic map (WT, blue; CNB+Latch, red) onto the PCA analysis plot.

DOI: https://doi.org/10.7554/eLife.44146.013

The following figure supplement is available for figure 5:

**Figure supplement 1.** Interactions between nucleotide coordinating elements (P Loop, Switch 1, Switch 2, and α0) in WT, CNB+Latch, and Latch mutant motors.

DOI: https://doi.org/10.7554/eLife.44146.014

---

of membrane-bound organelles and monitored transport to the cell periphery after 30 min (*Figure 6A*). Previous studies utilized single-particle tracking of peroxisomes in COS7 cells and found that they exhibit sub-diffusive motion in the perinuclear region and that 2–15 pN of force, depending on peroxisome size, is required to move a peroxisome away from this region (*Efremov et al., 2014*). Therefore, the peroxisome can be considered a low-load cargo (*Figure 6B*, *Schimert et al., 2019*) requiring teams of kinesin-1 motors to collectively transport against loads < 3 times greater than the force required to stall a single motor.

COS7 cells were cotransfected with a plasmid for expression of *Rn*KIF5C(1-560) motors tagged with monomeric NeonGreen (mNG) and FRB domain and a plasmid for expression of a peroxisome-targeted PEX-mRFP-FKBP fusion protein. In the absence of rapamycin, the PEX-RFP-FKBP fusion proteins localized to the peroxisome surface and the WT KIF5C-mNG-FRB proteins were diffusely localized throughout the cell (*Figure 6—figure supplement 1A*). Addition of rapamycin resulted in recruitment of motors to the peroxisome surface via dimerization of the FRB and FKBP domains and motor activity drove dispersion of the peroxisomes to the cell periphery (*Figure 6—figure supplement 1A*). Cargo dispersion before and after motor recruitment was quantified using two different

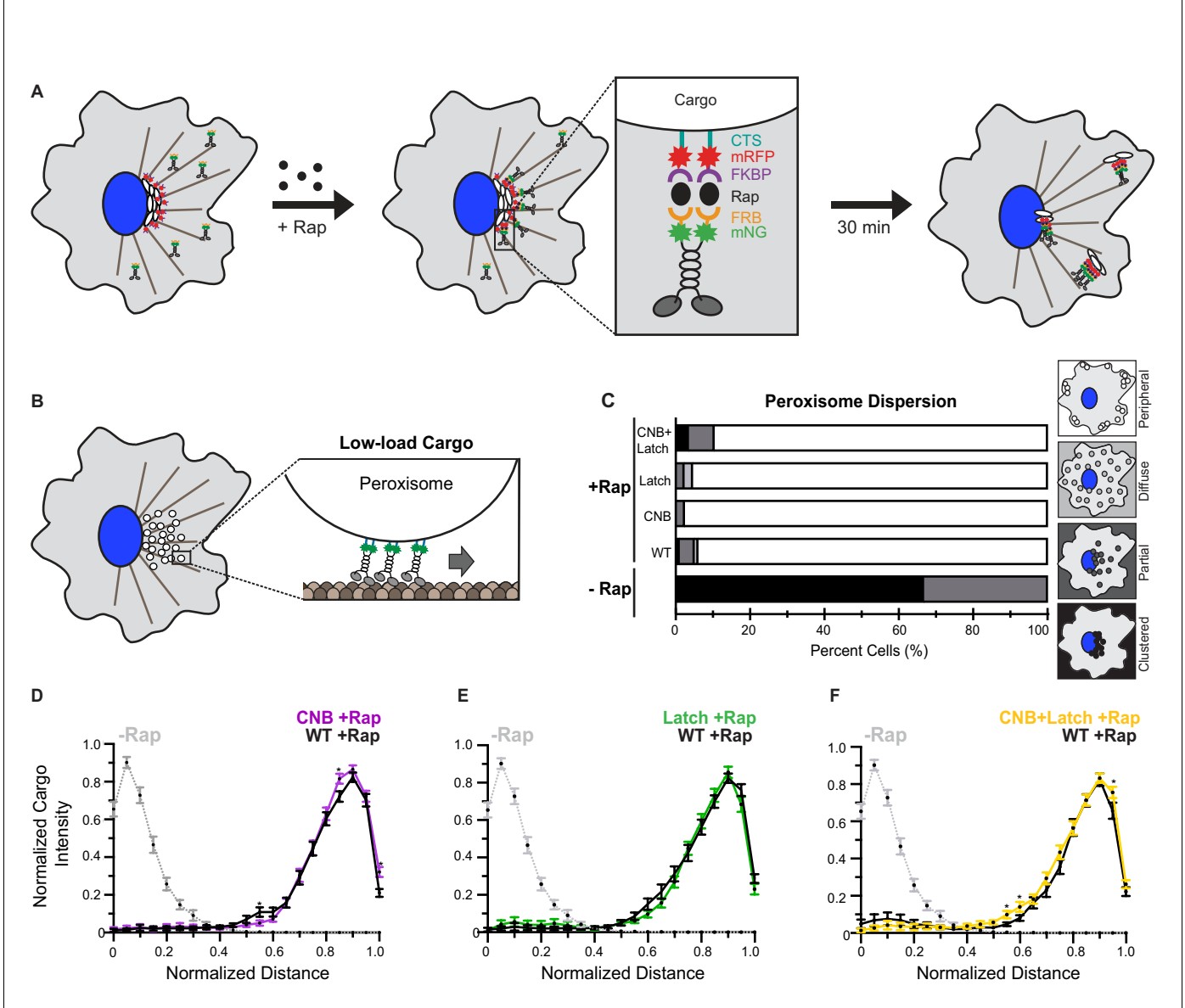

**Figure 6.** CNB and Latch mutations do not affect transport of peroxisomes (low-load cargo) by teams of kinesin-1 motors in cells. (**A**) Schematic of the inducible motor recruitment assay. A kinesin motor tagged with monomeric NeonGreen (mNG) and an FRB domain (KIF5C-mNG-FRB) is coexpressed with a cargo targeting sequence (CTS) tagged with monomeric red fluorescent protein (mRFP) and FKBP domain (CTS-mRFP-FKBP) in COS7 cells. Addition of rapamycin (+Rap) causes heterodimerization of the FRB and FKBP domains and recruitment of motors to the cargo membrane. Recruitment of active motors drives cargo dispersion to the cell periphery. (**B**) Schematic of the inducible peroxisome dispersion assay. Peroxisomes are loosely clustered in the perinuclear region of COS7 cells and are largely immotile, thus providing a low-load cargo for transport by teams of recruited motors. (**C**) Qualitative analysis of peroxisome dispersion. Peroxisome localization in individual cells was scored as clustered (black), partially dispersed (dark grey), diffusely dispersed (light grey), or peripherally dispersed (white) 30 min after recruitment of teams of WT, CNB, Latch, or CNB+Latch motors. The data for each construct is summarized as a stacked bar plot. For each construct, N ≥ 50 cells were analyzed across three separate experiments. Representative images of peroxisome dispersion by teams of WT or mutant motors are in ***Figure 6—figure supplement 1***. (**D–F**) Quantitative analysis of peroxisome dispersion. A radial profile of peroxisome intensity was generated for each cell and the data for each condition was converted to an averaged and normalized distance distribution across all cells. Each data point indicates the mean normalized cargo intensity ± SEM for N ≥ 50 cells across three separate trials. Gray dotted line: WT -Rap; Black line: WT +Rap; Purple line: CNB +Rap; Green line: Latch +Rap; Yellow line: CNB +Latch +Rap. Statistically significant differences in peroxisome localization comparing the mean normalized cargo intensity of the mutant motors to the wild type at any binned distance was determined; *, p<0.05. Detailed method of quantitative analysis of cargo dispersion is illustrated in ***Figure 6—figure supplement 2***.

DOI: https://doi.org/10.7554/eLife.44146.015

The following figure supplements are available for figure 6:

*Figure 6 continued on next page*

*Figure 6 continued*

**Figure supplement 1.** Peroxisome dispersion (low-load cargo) by teams of WT or CNB and/or NL docking mutant motors.
DOI: https://doi.org/10.7554/eLife.44146.016
**Figure supplement 2.** Analysis of cargo dispersion in cells.
DOI: https://doi.org/10.7554/eLife.44146.017

methods. First, cargo dispersion in each cell was qualitatively scored as clustered, partially dispersed, diffusely dispersed, or peripherally dispersed (*Figure 6—figure supplement 2A*) with the data for the population of cells summarized as a stacked bar plot. Second, peroxisome dispersion was quantified by generating a radial profile of cargo intensity for each cell and converting this profile into a normalized distance distribution; the distance distribution across multiple cells was then averaged across all cells for each motor construct (*Figure 6—figure supplement 2B*).

In the absence of rapamycin, peroxisomes were largely clustered in middle of cell (*Figure 6—figure supplement 1A,E*; *Figure 6C*, qualitatively 67% of cells have clustered peroxisomes; *Figure 6D*, quantitatively 95% of the peroxisome intensity adjacent to the nucleus). Thirty minutes after addition of rapamycin and recruitment of teams of WT kinesin-1 motors, peroxisomes were transported to the periphery of the cell (*Figure 6—figure supplement 1A,E*; *Figure 6C*, qualitatively 94% of cells have dispersed peroxisomes; *Figure 6D*, quantitatively 81% of the peroxisome intensity at the cell periphery). Notably, although mutant motors are crippled in their ability to transport against load as single motors in an optical trap (*Figure 3*), as teams these motors are able to cooperate to transport peroxisomes to the cell periphery to a similar extent as teams of WT motors. Specifically, thirty minutes after addition of rapamycin and motor recruitment, teams of CNB, Latch, or CNB+Latch mutant motors were able to disperse peroxisomes to the periphery of the cell (*Figure 6—figure supplement 1B–E*; *Figure 6C*, qualitatively 97%, 92%, and 97% of cells have dispersed peroxisomes, respectively; *Figure 6D–F*, quantitatively 84%, 81%, and 79% of the peroxisome intensity at the cell periphery, respectively). Statistical analysis indicates that peroxisome dispersion by the mutant motors was not significantly different than that of the WT motor (*Figure 6D–F*). These results suggest that impaired force generation by weakening CNB and/or N-latch formation can be overcome by teams of motors for efficient transport of a low-load, membrane-bound cargo in cells.

## CNB and N-latch mutations impair the ability of teams of kinesin-1 motors to transport high-load cargo (Golgi elements) in cells

Although CNB formation and NL latching were not required for teams of kinesin motors to transport peroxisomes to the cell periphery, we considered the possibility that NL docking may be critical under conditions where motors must generate high forces and work against high loads. To address how motors cooperate in teams to transport high-load cargo in cells, we used the inducible recruitment strategy (*Figure 6A*) to link teams of motors to the Golgi membrane using a GMAP210 Golgi-localization sequence (*Schimert et al., 2019*; *Nguyen et al., 2014*; *Engelke et al., 2016*) and monitored cargo transport to the cell periphery after 30 min. The Golgi is a compact organelle and its localization near the nucleus is maintained by a variety of mechanisms including microtubule minus-end directed activity of cytoplasmic dynein motors (*Brownhill et al., 2009*). Using an optical trap, Guet *et al.* determined that over 150 pN of force is required to deform the Golgi network in cells (*Guet et al., 2014*). Therefore, the Golgi can be considered a high-load cargo (*Figure 7A*, *Schimert et al., 2019*) as teams of motors driving Golgi dispersion are required to cooperate to transport against forces ~ 30 times greater than the force required to stall one kinesin-1 motor.

To validate that peroxisome and Golgi dispersion represent low- and high-load cargoes in cells, respectively, we assessed whether teams of kinesin-8 KIF18A motors, previously characterized to stall at 1 pN of force (*Jannasch et al., 2013*), can cooperate to transport peroxisomes and Golgi elements. Before addition of rapamycin, both peroxisomes and Golgi were clustered in middle of cell (*Figure 7—figure supplement 1A,B*). Thirty minutes after addition of rapamycin and recruitment of teams of KIF18A motors, peroxisomes were dispersed to the periphery of the cell (*Figure 7—figure supplement 1A–C*, qualitatively 91% of cells have dispersed peroxisomes, quantitatively 80% of peroxisome intensity at the cell periphery). However, thirty minutes after addition of rapamycin and recruitment of teams of KIF18A motors to Golgi membranes, a majority of Golgi elements remained localized in the perinuclear region of the cell with minimal accumulation at the cell periphery

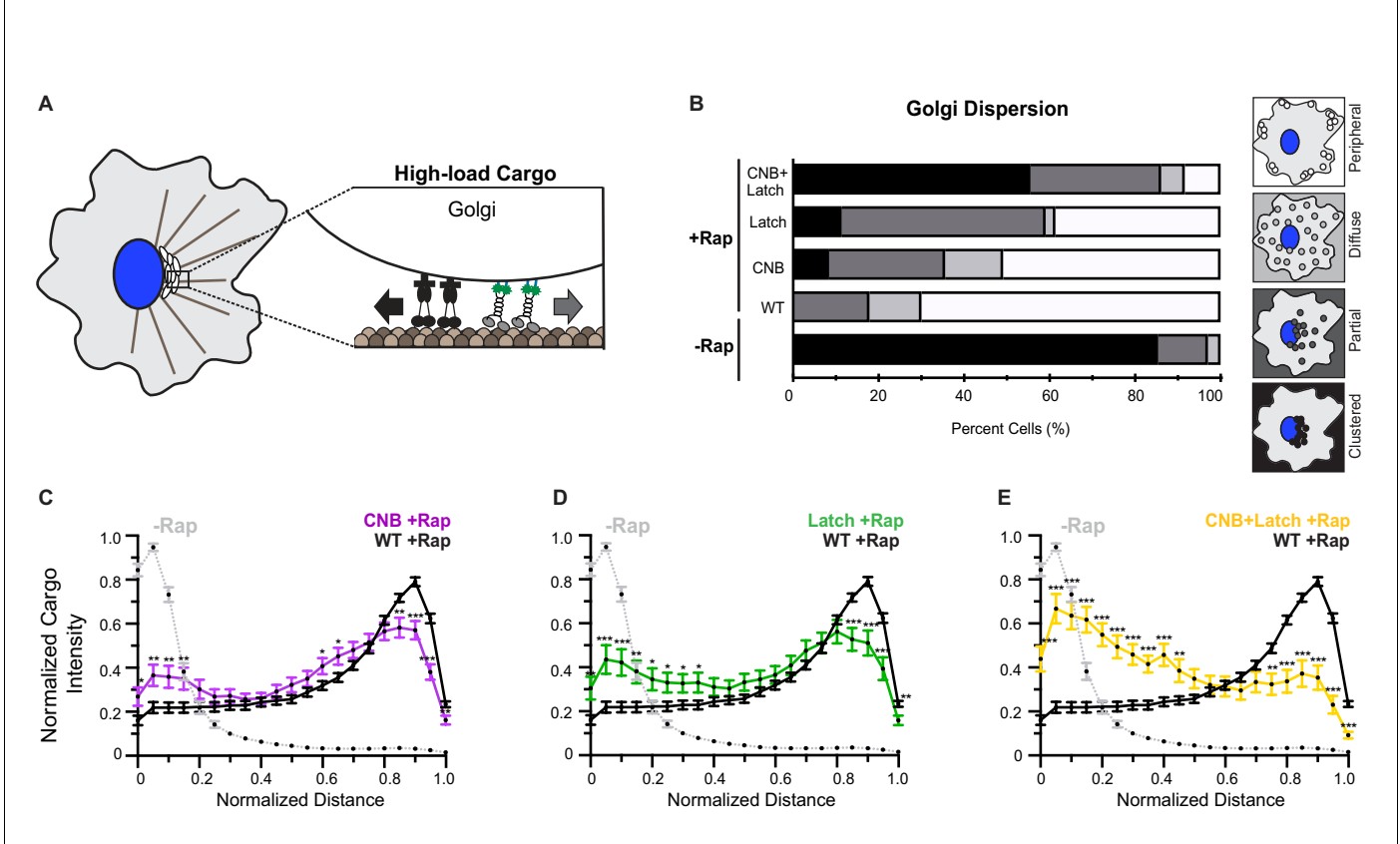

**Figure 7.** CNB and NL docking mutations impair transport of Golgi elements (high-load cargo) by teams of kinesin-1 motors in cells. (**A**) Schematic of inducible Golgi dispersion assay. A variety of mechanisms, including the action of cytoplasmic dynein motors (black), maintain the Golgi in a compact cluster near the nucleus. Thus, the Golgi is a high-load cargo for transport by teams of recruited team of kinesin motors. The ability of low force generating kinesin-8 motor to drive peroxisome and Golgi dispersion is shown in *Figure 7—figure supplement 1*. Representative images of Golgi dispersion be teams of WT, CNB, Latch, and CNB+Latch motors is shown in *Figure 7—figure supplement 2*. The ability of kinesin motors to drive Golgi dispersion after interfering with the function of endogenous dynein motors is shown in *Figure 7—figure supplement 3*. (**B**) Qualitative analysis of Golgi dispersion. COS7 cells were co-transfected with plasmids encoding for the expression of WT or mutant KIF5C-mNG-FRB motors together with the Golgi-targeting GMAP210p-mFRP-FKBP fusion protein. Motor recruitment was induced by addition of rapamycin (+Rap) and cells were fixed after 30 min and stained with an antibody to the Golgi marker giantin. Golgi localization in individual cells was scored as clustered (black), partially dispersed (dark grey), diffusely dispersed (light grey), or peripherally dispersed (white) after recruitment of teams of WT, CNB, Latch, or CNB+Latch motors. For each construct, N ≥ 30 cells were analyzed across three separate experiments. The results for each construct are summarized as a stacked bar plot. (**C**–**E**) Quantitative analysis of Golgi dispersion. A radial profile of Golgi intensity was generated for each cell and the data for each condition were converted to an averaged and normalized distance distribution across all cells. Each data point indicates the mean normalized cargo intensity ± SEM for N ≥ 30 cells across three separate experiments. Gray line: WT -Rap; Black line: WT +Rap; Purple line: CNB +Rap; Green line: Latch +Rap; Yellow line: CNB+Latch + Rap. Significant differences in mean normalized cargo intensity after recruitment of mutant motors as compared to WT motors are indicated for each distance; *, p<0.05; **, p<0.01; ***, p<0.001.

DOI: https://doi.org/10.7554/eLife.44146.018

The following figure supplements are available for figure 7:

**Figure supplement 1.** Validation of peroxisome and Golgi as low- and high-load cargoes, respectively.

DOI: https://doi.org/10.7554/eLife.44146.019

**Figure supplement 2.** Golgi dispersion (high-load cargo) by teams of WT or NL docking mutant motors.

DOI: https://doi.org/10.7554/eLife.44146.020

**Figure supplement 3.** Kinesin-1 CNB and/or Latch mutants can drive transport of reduced-load Golgi elements.

DOI: https://doi.org/10.7554/eLife.44146.021

(*Figure 7—figure supplement 1D–F*, qualitatively 86% of cells have clustered Golgi; quantitatively 66% of Golgi intensity adjacent to the nucleus). Overall these findings suggest that teams of KIF18A motors are able to generate sufficient force to transport peroxisomes but not Golgi elements to the cell periphery.

We thus examined the ability of teams of WT or mutant kinesin-1 motors to transport Golgi elements to the cell periphery. Before addition of rapamycin, the Golgi was clustered near the nucleus of the cell (*Figure 7—figure supplement 2A,E*; *Figure 7B*, qualitatively 85% of cells have clustered Golgi; *Figure 7C*, quantitatively 83% of Golgi intensity near the nucleus). 30 min after addition of rapamycin and recruitment of teams of WT kinesin-1 motors to the Golgi surface, Golgi elements were efficiently dispersed to the cell periphery (*Figure 7—figure supplement 2A,E*; *Figure 7B*, qualitatively 82% of cells have dispersed Golgi; *Figure 7C*, 50% of Golgi intensity at the cell periphery). However, hindering either CNB or N-latch formation resulted in motors that were crippled in their ability to cooperate in teams to transport Golgi elements to the cell periphery. Thirty minutes after addition of rapamycin and recruitment of teams of CNB or Latch mutants, a significant fraction of Golgi elements remained clustered in the perinuclear region rather than accumulated at the cell periphery (*Figure 7—figure supplement 2B,C,E*; *Figure 7B*, qualitatively only 64% of cells have dispersed Golgi for CNB and 42% for Latch mutant; *Figure 7C*, quantitatively only 34% of Golgi intensity at the cell periphery for CNB and *Figure 7D*, only 35% for Latch mutant).

In this cellular assay, the effects of the CNB and N-latch mutations were additive as teams of CNB +Latch mutants were even more crippled in their capacity to cooperate and transport Golgi elements than the CNB and Latch mutant motors. Upon recruitment of CNB+Latch mutant motors, the majority of the Golgi elements remained clustered in the perinuclear region of the cell (*Figure 7—figure supplement 2D,E*; *Figure 7B*, qualitatively only 13% of cells have dispersed Golgi; *Figure 7E*, quantitatively only 22% of Golgi intensity at the cell periphery). Statistical analysis indicates that there is significant impairment of Golgi dispersion to the cell periphery after recruitment of mutant motors compared to WT motors (*Figure 7C–E*).

To verify that the inability of the CNB, Latch, and CNB+Latch mutant motors to drive Golgi dispersion was due to the increased load imposed by this cargo, we repeated the assay in cells where the contribution of cytoplasmic dynein to Golgi clustering was reduced. To do this, we overexpressed a truncated dynein intermediate chain 2 (IC2-N237) that acts in a dominant negative (DN) manner to block endogenous dynein function and causes partial dispersion of the Golgi complex (*Figure 7—figure supplement 3A*, *King et al., 2003*; *Blasius et al., 2013*). In cells overexpressing the dynein DN, transport of Golgi elements by teams of KIF18A motors was enhanced (*Figure 7—figure supplement 3B,D*; qualitatively 54% of cells have dispersed Golgi; quantitatively 37% of Golgi intensity at the cell periphery). These results indicate that cytoplasmic dynein contributes to the forces that teams of recruited kinesin motors must overcome to transport Golgi elements to the cell periphery. In cells overexpressing the dynein DN, the ability of the CNB, Latch, and CNB+Latch mutant motors to drive Golgi dispersion was also enhanced (*Figure 7—figure supplement 3B,E–G*, qualitatively 96% of cells have dispersed Golgi for WT, 88% for CNB mutant, 69% for Latch mutant, and 64% for CNB+Latch mutant). Taken together, these results suggest that while weak kinesin motors can cooperate for transport of a low-load membrane-bound cargo, they are unable to work in teams when faced with a high-load cargo in cells.

## Discussion

Studies at the single motor level have led to the hypothesis that CNB formation is the force-generating element for kinesin motors. Here we use molecular dynamics simulations, optical trapping, and single-molecule assays to show that both CNB and N-latch formation are critical for single kinesin-1 motors to transport against force. Weakening of either CNB or N-latch formation results in motors that do not stall under load and detach at low forces. Under unloaded conditions, mutant motors display improved motility properties due to allosteric effects of NL docking on the microtubule and nucleotide binding regions of the motor. We also use cellular assays to examine the contribution of CNB and N-Latch formation to the ability of motors to work in teams for transport of membrane-bound organelles. We find that both CNB formation and NL latching are critical for kinesin-1 motors to transport a high-load cargo in cells, even when the motors are working in teams.

### Impact of CNB and N-latch formation on kinesin-1 force generation

We used MD simulations to prioritize contacts whose mutation would weaken CNB formation (interaction of β9 with CS) and/or NL latching (interaction of N-latch and β10 with β7). Importantly, the ability of the CNB and/or Latch mutant motors to undergo processive motility was not impaired

(*Figure 4*), indicating that the mutations are tolerated by the motor when stepping under no load. Measurements of individual motors in an optical trap demonstrate that disruption of either CNB or N-latch formation resulted in motors unable to stall under load and more likely to detach when subjected to low forces (average detachment force ~1 pN, *Figure 3*). The effects of the CNB and N-latch mutations were not additive as weakening of both elements in the CNB+Latch mutant resulted in motors with similar behavior under load and similar detachment forces as the CNB and Latch mutant motors.

These results support the model that the CS plays a critical mechanical role in the force generation of kinesin-1 motors (*Rice et al., 1999*; *Hwang et al., 2008*; *Khalil et al., 2008*; *Hesse et al., 2013*). It is interesting that mutation of the two CS residues (A5 and S8) in our rat kinesin-1 motor resulted in an average detachment force of ~1 pN whereas previous work demonstrated that mutation of the same CS residues in the fly kinesin-1 motor resulted in an average detachment force of ~3 pN (*Khalil et al., 2008*). An intriguing possibility is that the additional residues at the N-terminus of the fly kinesin-1 motor (*Figure 1C*) can engage in additional interactions with the core motor domain and thereby partially compensate for the mutations during CNB formation.

Our results provide the first experimental demonstration of the contribution of N-latch formation to kinesin-1 force generation. A role for N-latch formation was previously implicated in MD simulations of forced breakage of the N-latch which resulted in complete undocking of the NL (*Hwang et al., 2008*). Experimentally, a previous study mutated the N-latch of human kinesin-1 KIF5B (N332A) and found that force generation was crippled (*Rice et al., 2003*), however, the contribution of the N-latch in this study was examined in the context of i) mutation of the proceeding residue (V331A) predicted to be important for CNB formation, and ii) a cysteine-light kinesin-1 motor domain whose force dependence is dampened compared to WT kinesin-1 (*Andreasson et al., 2015*). Our results indicate that without N-latch formation, single kinesin-1 motors can generate a power stroke via CNB formation but cannot sustain force generation (*Figure 3*). At present, it is unclear whether an impaired N-latch disrupts force production due to dissociation of the CNB, as predicted in the simulations (*Khalil et al., 2008*), and/or due to an inability to coordinate processive stepping and the out-of-phase mechanochemical cycles of a dimeric kinesin-1 motor.

## Allosteric effects of CNB and N-latch mutations on unloaded motility

Although mutant motors with weakened NL docking underwent premature detachment under loaded conditions, they exhibited enhanced motility properties under unloaded conditions. Specifically, the CNB, Latch, and CNB+Latch mutant motors displayed enhanced velocity in single-molecule motility assays (*Figure 4*), similar to previous results with the CS mutant version of fly kinesin-1 (*Khalil et al., 2008*). The enhanced velocity is likely due to allosteric effects of NL docking on core motor regions that coordinate and bind nucleotide (α0, S1, S2, PL; *Figure 5B*) and could result in enhanced catalytic site closure that would favor ATP hydrolysis in the mutant motors. Our findings are consistent with previous structural and enzymatic studies suggesting that NL docking allosterically gates ATPase activity (*Hahlen et al., 2006*; *Cao et al., 2014*). Our findings are also consistent with previous time-resolved (TR)[2]FRET studies of kinesin-1 and kinesin-5 motors demonstrating that NL docking is allosterically coupled to active site closure (*Muretta et al., 2018*; *Muretta et al., 2015*). Collectively, these results highlight how subtle changes in NL docking elements (CS, β9, N-latch/β10) can act as a molecular gearshift, where speed and processivity comes at the cost of robust force production.

The CNB, Latch, and CNB+Latch mutant motors also displayed longer runs and an increased landing rate (*Figure 4*). These results indicate that NL docking also has allosteric effects on the microtubule-binding surface of kinesin-1. PCA analysis demonstrated that the major structural difference between motors in the ADP-bound and microtubule-free state and motors in the ATP-bound and microtubule-bound state is the positioning of α4 (*Figure 5C*). MD simulations of CNB+Latch mutant motors indicated an increased ability of this motor's microtubule-binding surface to sample a conformation compatible with strong microtubule binding (*Figure 5C*, α4 up). We suggest that these allosteric changes can account for the mutant motors' enhanced run lengths and ability to enter or re-enter a processive run.

That weakening of CNB and N-latch formation resulted in increased detachment under loaded conditions but reduced detachment under unloaded conditions is intriguing and further work is required to understand the mechanistic basis. These results highlight the benefits of a simulation-

guided approach to weaken single contact points between the NL and the core motor domain in a manner that can reveal mechanical features of the motor under load yet can be tolerated by the motor when stepping under no load. In the future, this approach can be utilized to examine how sequence changes in CS and NL elements across the kinesin superfamily impact the force generation and motility properties of these motors.

## Physiological relevance of NL docking and implications for multi-motor transport in cells

Our results suggest that both CNB formation and NL latching are essential for teams of kinesin-1 motors to collectively transport high-load, but not low-load, membrane-bound cargoes in cells. Teams of CNB and/or N-latch mutant motors were impaired in their ability to transport Golgi elements to the cell periphery as compared to WT kinesin-1 motors (*Figure 7*) but were able to transport peroxisomes to the cell periphery in a manner indistinguishable from the WT motor (*Figure 6*). Notably, this is the first evidence that a power stroke mechanism and force generation are critical for multi-motor driven transport under physiological conditions. Further support for the conclusion that force generation by individual kinesin-1 motors within a team is required for transport of high-load cargoes in cells comes from two additional findings. First, the kinesin-8 motor KIF18A, which generates only 1 pN of force individually (*Jannasch et al., 2013*), is able to cooperate to drive peroxisome motility but is unable to drive Golgi dispersion (*Figure 7—figure supplement 1*). Second, single-headed kinesin motors that weakly engage with the microtubule track as individual motors can cooperate to drive peroxisome motility but are largely unable to drive Golgi dispersion (*Schimert et al., 2019*). Taken together, these results suggest that motility properties other than a power stroke-like mechanism can contribute to collective motility and cargo transport but only under low-load conditions.

Interestingly, the CNB, Latch, and CNB+Latch mutant motors differ in their ability to work collectively to drive dispersion of a high-load cargo. Teams of CNB+Latch mutant motors were more impaired in their ability to drive Golgi dispersion as compared to the CNB and N-Latch mutants (*Figure 7*). This result was surprising as we observed similar properties under load for the CNB, Latch, and CNB+Latch mutants as single motors in optical trap assays (*Figure 3*). These results highlight the limitations of extrapolating single-molecule properties to understand motor behavior in teams and/or in a cellular environment. The most likely explanation for why the CNB+Latch mutant is more impaired in Golgi dispersion relates to differences in the assays themselves. The optical trap assay measures the behavior of single motors whereas the cargo dispersion assays report on the cooperative activity of teams of motors. In addition, in the optical trap assay, motors are statically attached to a bead whereas in the cellular assays, motors can freely diffuse in the lipid environment of the membrane-bound organelle.

In multi-motor assays, the motility of individual motors can be facilitated or hindered by other motors in the complex. For example, motors at the leading edge of a membrane-bound vesicle are predicted to shoulder most of the load during transport, and can generate assisting forces that promote the stepping of motors at the trailing edge (*Leduc et al., 2010*; *Nelson et al., 2014*). For kinesin-1, assisting loads as small as 1.5 pN could restore forward stepping and processive motion of a severely crippled kinesin-1 motor (*Khalil et al., 2008*). It may be that the CNB and Latch mutants are more amenable to assisting forces in the Golgi dispersion assay than the CNB+Latch mutant motors. Other parameters that have been noted to influence the ability of motors to work in teams are the load-dependent detachment of the motor from the microtubule track (*Arpağ et al., 2014*; *Norris et al., 2014*) and the ability to rebind to the microtubule after detachment (*Feng et al., 2018*). These parameters seem less likely to explain the differences between the CNB, Latch, and CNB+Latch mutant motors in Golgi dispersion as all of the mutant motors readily detached from the microtubule as single motors under load (*Figure 3*) and displayed an enhanced microtubule landing rates under unloaded conditions (*Figure 4*).

Although our findings provide strong support for the hypothesis that CNB formation, as the mechanical element for kinesin force production, is required for high-load cargo transport in cells, it is possible that the differences observed between the peroxisome and Golgi dispersion assays are due to experimental conditions rather than motor force generation. While experimental conditions such as kinesin expression level and the effectiveness of motor recruitment have minimal influence on peroxisome dispersion in these assays (*Kapitein et al., 2010*; *Efremov et al., 2014*;

*Schimert et al., 2019*), it is possible that differences in size and/or local microenvironment fluidity result in cargo-specific steric effects and/or drag forces that impact transport driven by the exogenous motors. Further work with reconstitution systems are required to examine these possibilities.

## Implications for CNB formation and NL docking in other kinesins

The motor domain is highly conserved in both sequence and structure across the kinesin superfamily and many chemical and mechanical features are likely to be shared across all members. Formation of a CNB has also been observed structurally for members of the kinesin-3, kinesin-5 and kinesin-6 families (*Atherton et al., 2017*; *Ren et al., 2018*; *Hesse et al., 2013*) although a mechanical role in force generation has only been tested for kinesin-1 motors. Whether N-latch formation and docking of β10 along the core motor domain play important roles beyond kinesin-1 remain to be investigated. The asparagine residue involved in N-latch formation is highly conserved across processive kinesins with the exception of kinesin-6 family members (*Figure 1—figure supplement 2*), and recent work failed to resolve a docked NL conformation for the kinesin-6 motor MKLP2 (human KIF20A) even in the ATP- and microtubule-bound state (*Atherton et al., 2017*).

In the absence of ATP, formation of the CNB is prevented by occupancy of the docking pocket by the hydrophobic CTR (I9 in kinesin-1 KIF5C [*Sindelar and Downing, 2010*; *Shang et al., 2014*; *Nitta et al., 2008*; *Cao et al., 2014*; *Sindelar, 2011*]). Structural changes induced by ATP binding open up this pocket to occupancy by the NIS (I327 in kinesin-1 KIF5C [*Sindelar and Downing, 2010*; *Shang et al., 2014*; *Nitta et al., 2008*; *Cao et al., 2014*; *Sindelar, 2011*). As an isoleucine or valine residue is found in the CTR and NIS positions across a large number of kinesin sequences (*Figure 1—figure supplement 2*), mutually exclusive access of the CTR and the NIS to the docking cleft may be a shared feature for N-kinesins that generate force and processive motility. In support of this possibility, structural studies have demonstrated that the docking pocket is occluded by CS residues in the absence of ATP for kinesin-3 and kinesin-6 motors (*Atherton et al., 2017*; *Arora et al., 2014*).

How variations in the length and sequence of the CS influence family-specific force and motility properties is not understood. We note that kinesin-3 motors have the shortest CS (*Figure 1—figure supplement 2*) and a recent study found that the kinesin-3 motor KIF13B forms a short CNB with weaker CS-NL interactions than kinesin-1 (*Ren et al., 2018*). Given our results, a short CS and weak CNB could contribute to the fast and superprocessive motility and high landing rate observed for motors in the kinesin-3 family (*Soppina et al., 2014*; *Soppina and Verhey, 2014*) as well as their tendency to detach from the microtubule under load (*Arpağ et al., 2014*; *Norris et al., 2014*). Interestingly, a short CS and weak CNB formation do not appear to negatively impact force output as single kinesin-3 motors are capable of withstanding forces equivalent to that of single kinesin-1 motors (~6 pN, [*Huckaba et al., 2011*; *Tomishige et al., 2002*]). Further work on the force generation of kinesin-3 and other family members will provide important information about mechanical and structural features shared across the kinesin superfamily.

## Materials and methods

### Key resources table

| Reagent type | Designation | Source or reference | Identifiers | Additional information |
|---|---|---|---|---|
| Cell line | COS-7 (*Cercopithecus aethiops*) male | AATC | Cat. #: CRL-1651, RRID: CVCL_0224 | |
| Transfected construct | KIF5C(1-560)-mNG-FRB | (*Schimert et al., 2019*) PMID: 30850543 | | |
| Transfected construct | PEX3-mRFP-FKBP | (*Kapitein et al., 2010*) PMID: 20923648 | | |
| Transfected construct | GMAP-mRFP-FKBP | (*Engelke et al., 2016*) PMID: 27045608 | | |
| Transfected construct | dyneinIC2-N237 | (*King et al., 2003*) PMID: 14565986 | | |

*Continued on next page*

*Continued*

| Reagent type | Designation | Source or reference | Identifiers | Additional information |
|---|---|---|---|---|
| Transfected construct | KIF18A(1-452) | (*Weaver et al., 2011*) PMID: 21885282 | | |
| Antibody | cis-Golgi marker giantin | BioLegend | Cat. #: 924302 RRID: AB_2565451 | IF(1:1200) |
| Antibody | β-tubulin | Developmental Studies Hybridoma Bank | Cat. #: E7, RRID: AB_528499 | IF(1:2000) |
| Antibody | goat anti-rabbit Alexa680 | Jackson Immuno Research Labs | Cat. #: 111-625-144, RRID: AB_2338085 | IF(1:500) |
| Antibody | goat anti-mouse Alexa350 | Molecular Probes | Cat. #: A-11045, RRID: AB_142754 | IF(1:500) |
| Drug | rapamycin | Calbiochem, Millipore Sigam | Cat. #: 553210 | |
| Software | GraphPad Prism | GraphPad Software Inc | RRID: SCR_002798 | Version 7.0 c |
| Software | MATLAB, code used for single molecule analysis | MathWorks | RRID: SCR_001622 | R2016b, PMID: 25365993 |
| Software | Origin 2017 | OriginLab | RRID: SCR_014212 | |
| Software | RStudio, code used for quantitative dispersion analysis | Rstudio | RRID: SCR_000432 | Version 3.4.1, manuscript in preparation |
| Software | ImageJ | NIH | RRID: SCR_003070 | |
| Software | PyMOL | PyMOL Molecular Graphics System, Schrödinger | RRID: SCR_000304 | Version 2.2.0 |
| Software | MODELLER | PMID: 8254673 | RRID: SCR_0083595 | Version 9.18 |
| Software | AMBER 14 Package | PMID: 16200636 | RRID: SCR_014230 | |

## Plasmids

A truncated, constitutively active kinesin-1 [rat KIF5C(1-560)] was used (*Cai et al., 2007*). Point mutations to impair CNB and/or N-latch formation were generated using QuickChange site-directed mutagenesis and all plasmids were verified by DNA sequencing. Motors were tagged with three tandem monomeric Citrine fluorescent proteins (3xmCit) for single molecule imaging assays (*Cai et al., 2007*), a FLAG-tag for optical trapping assays, and monomeric NeonGreen (mNG)-FRB for inducible cargo dispersion assays in cells (*Kapitein et al., 2010*). The peroxisome-targeting PEX3-mRFP-FKBP construct was a gift from Casper Hoogenraad (Utrecht University [*Kapitein et al., 2010*]). The Golgi-targeting GMAP-mRFP-FKBP construct is described in *Schimert et al. (2019)* and *Engelke et al. (2016)*. KIF18A(1-452) was a gift from Claire Walczak (Indiana University [*Weaver et al., 2011*]). Constructs coding for FRB (DmrA) and FKBP (DmrC) sequences were obtained from ARIAD Pharmaceuticals and are now available from Takara Bio Inc. Plasmids encoding monomeric NeonGreen were obtained from Allele Biotechnology.

## Cell culture, transfection, and lysate preparation

COS-7 (African green monkey kidney fibroblasts, American Type Culture Collection, RRID:CVCL_0224) were grown at 37°C with 5% (vol/vol) $CO_2$ in Dulbecco's Modified Eagle Medium (Gibco) supplemented with 10% (vol/vol) Fetal Clone III (HyClone) and 2 mM GlutaMAX (L-alanyl-L-glutamine dipeptide in 0.85% NaCl, Gibco). Cells are checked annually for mycoplasma contamination and were authenticated through mass spectrometry (the protein sequences exactly match those in the African green monkey genome). 24 hr after seeding, the cells were transfected with plasmids encoding for the expression of motor tagged with three tandem monomeric citrines or FLAG, TransIT-LT1 transfection reagent (Mirus), and Opti-MEM Reduced Serum Medium (Gibco). Cells were trypsinized and harvested 24 hr after transfection by low-speed centrifugation at 3000 x *g* at 4°C for 3 min. The

pellet was resuspended in cold 1X PBS, centrifuged at 3000 x $g$ at 4°C for 3 min, and the pellet was resuspended in 50 µL of cold lysis buffer [25 mM HEPES/KOH, 115 mM potassium acetate, 5 mM sodium acetate, 5 mM MgCl$_2$, 0.5 mM EGTA, and 1% (vol/vol) Triton X-100, pH 7.4] with 1 mM ATP, 1 mM phenylmethylsulfonyl fluoride, and 1% (vol/vol) protease inhibitor cocktail (P8340, Sigma-Aldrich). Lysates were clarified by centrifugation at 20,000 x $g$ at 4°C for 10 min and lysates were snap frozen in 5 µL aliquots in liquid nitrogen and stored at −80°C.

## Optical trapping assays

Tubulin was reconstituted and polymerized into microtubules as described previously (*Reinemann et al., 2017*; *Reinemann et al., 2018*). Tubulin (bovine brain, Cytoskeleton TL238) was reconstituted in 25 mL BRB80 buffer [80 mM PIPES (Sigma P-1851), 1 mM EGTA (Sigma E-4378), 1 mM MgCl2 (Mallinckrodt H590), pH adjusted to 6.9 with KOH] supplemented with 1 mM GTP (Cytoskeleton BST06) and kept on ice. 13 mL PEM104 buffer (104 mM PIPES, 1.3 mM EGTA, 6.3 mM MgCl2, pH adjusted to 6.9 with KOH), 2.2 mL 10 mM GTP, and 2.2 mL DMSO were mixed. 4.8 mL of 10 mg/mL tubulin were added to the mixture and allowed to incubate for 40 min at 37°C. Subsequently, 2 mL of stabilization solution [STAB: 38.6 mL PEM80, 0.5 mL 100 mM GTP, 4.7 mL 65 g/L NaN3 (Sigma S-8032), 1.2 mL 10 mM Taxol (Cytoskeleton TXD01), 5 mL DMSO (Cytoskeleton)] was added to the stock microtubule solution at room temperature.

Optical trap assays were performed as described previously (*Reinemann et al., 2017*; *Reinemann et al., 2018*). 0.44 µm anti-FLAG-coated beads were prepared by crosslinking anti-FLAG (Thermo Fisher Scientific) antibodies to carboxy polystyrene beads (Spherotech) via EDC chemistry. Lysates containing FLAG-tagged motors were diluted in assay buffer [AB: P12 buffer (12 mM PIPES (Sigma P-1851), 1 mM EGTA (Sigma E-4378), 1 mM MgCl2 (Mallinckrodt H590), pH adjusted to 6.9 with KOH), 1 mM DTT (Sigma Aldrich), 20 mM Taxol (Cytoskeleton), 1 mg/mL casein (Blotting-Grade Blocker, Biorad), 1 mM ATP (Sigma Aldrich)] were incubated with gently sonicated anti-FLAG beads to allow binding for 1 hr at 4°C on a rotator in the presence of oxygen scavenging reagents (5 mg/mL b-D-glucose (Sigma Aldrich), 0.25 mg/mL glucose oxidase (Sigma Aldrich), and 0.03 mg/mL catalase (Sigma Aldrich).

A flow cell that holds a volume of ~15 µL was assembled using a microscope slide, etched coverslips, and double-sided sticky tape. Before assembly, etched coverslips were incubated in a solution of 100 µL poly-l-lysine (PLL, Sigma P8920) in 30 mL ethanol for 15 min. The coverslip was then dried with a filtered air line. After flow cell assembly, microtubules were diluted 150 times from the stock in a solution of PemTax (1 µL 10 mM Taxol in 500 µL P12). The diluted microtubules were added to the flow cell and allowed to incubate to the PLL surface for 10 min. Unbound microtubules were then washed out with 20 µL PemTax. A solution of casein (Blotting-Grade Blocker, Biorad 1706404) diluted in PemTax (1:8 mixture) was then added to the flow cell and allowed to incubate for 10 min to block the remainder of the surface to prevent non-specific binding. After the incubation, the flow cell was washed with 50 µL PemTax and 80 µL assay buffer (AB). 20 µL of the bead/motor incubation was then added to the flow cell.

Optical trapping measurements were obtained using a custom-built instrument with separate trapping and detection systems. The instrument setup and calibration procedures have been described previously (*Khalil et al., 2008*). Briefly, beads were trapped with a 1,064 nm laser that was coupled to an inverted microscope with a 100x/1.3 NA oil-immersion objective. Bead displacements from the trap center were recorded at 3 kHz and further antialias filtered at 1.5 kHz. To ensure that we were at the single molecule limit for the motility assay, the protein-bead ratio was adjusted so that fewer than half of the beads trapped and tested on microtubules showed binding, actually having 5–10% binding the majority of the time. A motor-coated bead was trapped in solution and subjected to position calibration and trap stiffness Labview routines. Afterward, the bead was brought close to a surface-bound microtubule to allow for binding. Bead position displacement and force generation were measured for single motor-bound beads. Detachment forces are plotted as a dot plot where each dot indicates the maximum detachment force of an event and the mean for each construct is indicated by a black horizontal line. Statistical differences between the maximum detachment force of wild type and mutant motors were calculated by using a two-tailed unpaired Student's *t* test.

## Single-molecule motility assays

Microtubules were polymerized (purified tubulin unlabeled and HiLyte-647-labeled tubulin, Cytoskeleton Inc) in BRB80 buffer (80 mM Pipes/KOH pH 6.8, 1 mM $MgCl_2$, 1 mM EGTA) supplemented with GTP and $MgCl_2$ and incubated for 60 min at 37° C. 2 mM taxol in prewarmed BRB80 was added and incubated for 60 min to stabilize microtubules. Microtubules were stored in the dark at room temperature for up to 2 weeks. Flow cells were prepared by attaching a #1.5 $mm^2$ coverslip (Thermo Fisher Scientific) to a glass slide (Thermo Fisher Scientific) using double-sided tape. Microtubules were diluted in fresh BRB80 buffer supplemented with 10 µM taxol, infused into flow cells, and incubated for four minutes to allow for nonspecific absorption to the glass. Flow cells were then incubated with blocking buffer [30 mg/mL casein in P12 buffer (12 mM Pipes/KOH pH 6.8, 1 mM $MgCl_2$, 1 mM EGTA) supplemented with 10 µM taxol] for four minutes. Flow cells were then infused with motility mixture (0.5–1.0 µL of COS7 cell lysate, 25 µL P12 buffer, 15 µL blocking buffer, 1 mM ATP, 0.5 µL 100 mM DTT, 0.5 µL, 0.5 µL 20 mg/mL glucose oxidase, 0.5 µL 8 mg/mL catalase, and 0.5 µL 1 M glucose), sealed with molten paraffin wax, and imaged on an inverted Nikon Ti-E/B total internal reflection fluorescence (TIRF) microscope. with a perfect focus system, a 100 × 1.49 NA oil immersion TIRF objective, three 20 mW diode lasers (488 nm, 561 nm, and 640 nm) and EMCCD camera (iXon$^+$ DU879; Andor). Image acquisition was controlled using Nikon Elements software and all assays were performed at room temperature.

Motility data were analyzed by first generating maximum intensity projections to identify microtubule tracks (width = 3 pixels) and then generating kymographs in ImageJ (National Institutes of Health). Only motility events that lasted for at least three frames were analyzed. Furthermore, events that ended as a result of a motor reaching the end of a microtubule were included; therefore, the reported run lengths for highly processive motors are likely to be an underestimation. For the Latch and CNB+Latch motors, the run lengths are reported as the distance moved between gaps in the runs. Run length and velocities were plotted as cumulative distributions in MATLAB and used for statistical analysis. The cumulative distributions of motor velocities were fit to a Gaussian cumulative distribution as previously described (*Arpağ et al., 2014*; *Norris et al., 2014*) and a one-way analysis of variance test was used to assess whether velocity distributions were significantly different between motors. The cumulative distribution of WT motor run lengths was fit to an exponential distribution as previously described (*Norris et al., 2014*). However, a fit to an exponential decay function was not an appropriate model to describe the cumulative distributions of the CNB, Latch, and CNB +Latch motor run lengths. Rather, the cumulative distributions of the run lengths of the mutant motors were well fit to a gamma distribution. The scale parameter was fixed (assuming a rate of 1 or 2) and the shape parameter was the only fit parameter. The expected mean run length was calculated by multiplying the shape and scale parameters. A Kuskal-Wallis one-way analysis of variance was used to assess whether run length distributions were significantly different between motors. For each motor construct, the velocities and run lengths were binned and a histogram was generated by plotting the number of motility events for each bin. A corresponding Gaussian, exponential, or gamma distribution was overlaid on each histogram plot using rate and shape parameters derived from fitting the cumulative distributions.

## Molecular dynamics simulations

Simulation models of rat kinesin-1 (*Rn*KIF5C) motor domain in complex with tubulin heterodimer were constructed for motors in the no nucleotide (apo) and ATP-bound states based on PDB 4LNU (*Cao et al., 2014*) and PDB 4HNA (*Gigant et al., 2013*), respectively. Since the motor domain in both template structures (PDBs 4LNU and 4HNA) is KIF5B, residues that differ were mutated to match the sequence of rat KIF5C (UniprotID: P56536). The tubulin dimer was left unmodified. Missing coordinates were modeled using MODELLER v9.18 (*Sali and Blundell, 1993*). The ATP-hydrolysis transition-state analog, ADP–$AlF_4^-$, in PDB 4HNA was converted to ATP. The resulting systems of motor domain associated with tubulin dimer contained a total of ~170,000 atoms each. Models of ADP-bound wildtype and CNB+Latch mutant motor domains (not associated with the tubulin heterodimer) were prepared from PDB 2KIN (*Sack et al., 1997*).

Energy minimization and molecular dynamics simulations were performed with AMBER14 (*Case et al., 2005*) and the ff99SB AMBER force field (*Hornak et al., 2006*). Nucleotide parameters were obtained from *Meagher et al. (2003)*. Histidine protonation states were assigned based on

the their pKa values calculated by PROPKA (*Bas et al., 2008*). Starting structures were solvated in a cubic box of pre-equilibrated TIP3P waters molecules, extending 12 Å in each dimension from the surface of the solute. Sodium ions ($Na^+$) were added to neutralize the systems, followed by addition of sodium and chloride ($Cl^-$) ions to bring the ionic strength to 0.050 M. Energy minimization was performed in four stages, with each stage consisting of 500 steps of steepest descent followed by 4000 steps of conjugate gradient. First, minimization of solvent was performed by keeping positions of protein and nucleotides fixed. Second, side-chains and nucleotides were relaxed keeping the backbone positions fixed. Third, protein and nucleotide atoms were relaxed while keeping the solvent atoms fixed. Fourth, a last minimization stage was performed with no restraints. The system was gradually heated to 300K over 25 ps of simulation time in constant-volume (NVT) and periodic boundary conditions (PBC), with restraint of 20 kcal/mol/Å$^2$ on backbone atoms. Constant-temperature (T = 300K) and constant-pressure (p=1 bar) (NpT) equilibration was then performed in six stages. First, a 400 ps NpT equilibration was performed with restraint of 20 kcal/mol/Å$^2$ on backbone atoms. Further stages involved gradually reducing restraints of 20, 10, 5, and one kcal/mol/Å$^2$ on α carbons over five ns each. A final NpT equilibration was carried out without any restraints for five ns. Subsequent production phase molecular dynamics simulations were then performed under NpT and PBC with random velocity assignments for each run. Particle-mesh Ewald summation was adopted for treating long-range electrostatics. A 12 Å cutoff for energy minimization, and a 10 Å cutoff for molecular dynamics simulations was used to truncate non-bonded interactions. A two fs time-step was adopted for all molecular dynamics simulations. Hydrogen atoms were constrained using the SHAKE algorithm. All simulations were performed in-house on NVIDIA GPU cards with the GPU version of PMEMD (pmemd.cuda). Molecular dynamics simulations were started from equilibrated structures with four independent runs of 100 ns each. Trajectory analysis was carried out in R using the Bio3D v2.3–3 package (*Grant et al., 2006*; *Skjærven et al., 2014*).

## Simulation analysis: inter-residue distances

Statistically significant residue-residue distance differences between apo, ATP-bound and mutant states were identified with ensemble difference distance matrix (eDDM) analysis routine (*Muretta et al., 2018*). A total of 400 conformations were obtained for each state under comparison by extracting 100 equally time-spaced conformations from the last 20 ns of each simulation replicate. Distance matrices for each state were constructed from residue-residue distances, defined as the minimum distance between all heavy atoms of every residue pair in a given conformation. The distances matrices were processed by applying a smooth function to mask long distances as follows:

$$f(x) = \begin{cases} x, & x \leq c1 \\ c1 + 2*\left(\frac{c2-c1}{\pi}\right)*\cos\left(\frac{\pi}{2}*\frac{c2-x}{c2-c1}\right), & c1 < x \leq c2 \\ c1 + 2*\left(\frac{c2-c1}{\pi}\right), & x > c2 \end{cases} \tag{1}$$

where $x$ is residue-residue distance, $c1$ and $c2$ are parameters of the smooth function, set to 4 Å and 8 Å, respectively. The above routine reduces the difference between long distances while difference between short distances are kept intact. The significance of residue distance variation between apo and ATP-bound states, and between ATP-bound and mutant states, were evaluated with the Wilcoxon test. Residue pairs showing a p-value<0.05 and an average masked distance difference >1 Å were considered statistically significant residue-residue distance differences for further analysis (*Supplementary files 1*, *2,* and *3*).

## Simulation analysis: principal component analysis

A set of 17 experimental structures from the RCSB protein data bank, nine in ADP-like state not associated with the microtubule and eight in ATP-like state bound to tubulin heterodimer, were selected for examining the major conformational differences of the kinesin motor domain in these two states (*Supplementary file 4*). Principal component analysis (PCA) is a dimensionality reduction technique involving orthogonal transformation of the original data into a set of linearly uncorrelated variables termed principal components. Briefly, PCA involves diagonalization of the covariance matrix *C*, whose elements *Cij* are calculated from the Cartesian coordinates of Cα atoms, *r*, after superposition:

$$C_{ij} = \langle\, (r_i - \langle r_i \rangle) \cdot (r_j - \langle r_j \rangle)\, \rangle \qquad\qquad (2)$$

where $i$ and $j$ represent all pairs of 3N coordinates. The eigenvectors, or principal components (PCs), of the covariance matrix form a linear basis set of the distribution of structures. The variance of the distribution along each eigenvector is given by the corresponding eigenvalue. Projecting structures onto a sub-space defined by principal components with the largest variance (largest eigenvalues) provides a lower dimensional representation of the structure dataset.

PCA was performed on 112 equivalent, non-gap Cα atoms from each of the structures after superposition onto an invariant core comprising of structural elements β1, β2, β3, P-loop, α2, β6, β7 and α6 (*Scarabelli and Grant, 2013*). The trajectories from MD simulations of ADP-bound wildtype and CNB+Latch mutant kinesin motor domains were projected on to the PC sub-space defined by the first two PCA eigenvectors to allow comparison of the conformational space spanned by the simulations and the experimental structures (*Figure 4C*).

### Inducible cargo dispersion assays

Plasmids for expression of wild type or mutant Kif5C(560) motors tagged with monomeric Neon-Green and an FRB domain were cotransfected into COS-7 cells with a plasmid for expression of PEX3-mRFP-FKBP or GMAP210p-mRFP-2xFKBP at a ratio of 6:1 and 3:1 respectively with TransIT-LT1 transfection reagent (Mirus). Eight hours after transfection, rapamycin (Calbiochem, Millipore Sigma) was added to final concentration of 44 nM to promote FRB and FKBP heterodimerization and recruitment of motor the peroxisome or Golgi surface. Cells were fixed with 3.7% formaldehyde (Thermo Fisher Scientific) in 1X PBS, quenched in 50 mM ammonium chloride in PBS for 5 min, permeabilized for 5 min in 0.2% Triton-X 100 in PBS for 5 min and blocked in 0.2% fish skin gelatin in PBS for 5 min. Primary and secondary antibodies were added to blocking buffer and incubated for 1 hr at room temperature. Primary antibodies: polyclonal antibody against cis-Golgi marker giantin (1:1200 PRB-114C, Covance), antibody against β-tubulin (1:2000, Developmental Studies Hybridoma Bank #E7). Secondary antibodies: goat anti-rabbit Alexa680-labeled secondary antibody (1:500, Jackson ImmunoResearch). Coverslips were mounted in ProlongGold (Invitrogen) and imaged using an inverted epifluorescence microscope (Nikon TE2000E) with a 40 × 0.75 NA objective and a Cool-SnapHQ camera (Photometrics). Only cells expressing low levels of motor-mNG-FRB were imaged and included in quantification, as high expression of mutant KIF5C disrupted the microtubule network. Cargo localization before and after motor recruitment was quantified using two different methods. First, the phenotype of cargo dispersion was scored as clustered, partial dispersion, diffuse, or peripheral dispersion based on the signal localization in the PEX3 (peroxisome) or giantin (Golgi) signal. Second, a distance-based analysis using a custom ImageJ plugin was applied (*Figure 6—figure supplement 2*, manuscript in preparation). Statistical differences between mean cargo intensity at each binned distance between wild type and mutant motors were calculated by using a two-tailed unpaired Student's *t* test.

## Acknowledgements

We gratefully acknowledge Will Hancock for supplying purified KIF5C(1-560). This work was supported by a grant from NIH/NIGMS to KJV and BJG (R01GM070862) and a grant from NSF to MJL (NSF grant no. 1330792). BB was supported by a Graduate Research Fellowship from the National Science Foundation under grand no. DGE 1256260 and by the NIH Cellular and Molecular Biology Training Grant T32-GM007315. SJ was supported by the Qatar Research Leadership Program and by the Endowment for Basic Sciences at the University of Michigan. DNR was supported by the National Science Foundation Graduate Research Fellowship Program under grant no. 1445197.

# Additional information

## Funding

| Funder | Grant reference number | Author |
|--------|------------------------|--------|
| National Institutes of Health | R01GM070862 | Barry J Grant<br>Kristen J Verhey |
| National Science Foundation | 1330792 | Matthew J Lang |
| Qatar Leadership Program | | Shashank Jariwala |
| National Science Foundation | 1256260 | Breane G Budaitis |
| National Science Foundation | 1445197 | Dana N Reinemann |
| National Institutes of Health | T32GM007315 | Breane G Budaitis |

The funders had no role in study design, data collection and interpretation, or the decision to submit the work for publication.

## Author contributions
Breane G Budaitis, Conceptualization, Software, Formal analysis, Validation, Investigation, Visualization, Methodology, Writing—original draft, Project administration, Writing—review and editing; Shashank Jariwala, Software, Formal analysis, Validation, Investigation, Visualization, Methodology, Writing—review and editing; Dana N Reinemann, Formal analysis, Validation, Investigation, Visualization, Writing—review and editing; Kristin I Schimert, Formal analysis, Validation, Investigation, Visualization, Methodology, Writing—review and editing; Guido Scarabelli, Software, Formal analysis, Investigation; Barry J Grant, Conceptualization, Resources, Software, Formal analysis, Supervision, Funding acquisition, Investigation, Visualization, Writing—review and editing; David Sept, Resources, Software, Formal analysis, Supervision, Validation, Investigation, Methodology, Writing—review and editing; Matthew J Lang, Conceptualization, Resources, Software, Supervision, Funding acquisition, Writing—review and editing; Kristen J Verhey, Conceptualization, Resources, Formal analysis, Supervision, Funding acquisition, Writing—original draft, Project administration, Writing—review and editing

## Author ORCIDs
Kristin I Schimert (iD) http://orcid.org/0000-0001-9209-7986
Kristen J Verhey (iD) https://orcid.org/0000-0001-9329-4981

## Decision letter and Author response
Decision letter https://doi.org/10.7554/eLife.44146.028
Author response https://doi.org/10.7554/eLife.44146.029

# Additional files

## Supplementary files
• Supplementary file 1. List of residue-residue distances for WT KIF5C in the apo versus ATP-bound states. Differences in residue-residue distances based on MD simulations of tubulin-bound motors in the ATP-bound, post power stroke state.
DOI: https://doi.org/10.7554/eLife.44146.022

• Supplementary file 2. List of residue-residue distances for WT versus Latch mutant motors in the tubulin- and ATP-bound states. Differences in residue-residue distances are based on MD simulations of tubulin-bound motors in the ATP-bound, post power stroke state.
DOI: https://doi.org/10.7554/eLife.44146.023

• Supplementary file 3. List of residue-residue distances for WT versus CNB+Latch mutant motors in the tubulin- and ATP-bound states. Differences in residue-residue distances are based on MD simulations of tubulin-bound motors in the ATP-bound, post power stroke state.
DOI: https://doi.org/10.7554/eLife.44146.024

• Supplementary file 4. List of structures used for PCA analysis.
DOI: https://doi.org/10.7554/eLife.44146.025
• Transparent reporting form
DOI: https://doi.org/10.7554/eLife.44146.026

## Data availability

All data generated or analyzed during this study are included in the manuscript and supporting files.

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
