## [Decision Letter]

Thank you for submitting your article "Neck linker docking is critical for Kinesin-1 force generation in cells but at a cost to motor speed and processivity" for consideration by *eLife*. Your article has been reviewed by three peer reviewers, one of whom is a member of our Board of Reviewing Editors, and the evaluation has been overseen by a Reviewing Editor and Anna Akhmanova as the Senior Editor. The following individuals involved in review of your submission have agreed to reveal their identity: Ahmet Yildiz (Reviewer #2); William O Hancock (Reviewer #3).

The reviewers have discussed the reviews with one another and the Reviewing Editor has drafted this decision to help you prepare a revised submission.

Summary:

The authors use a variety of complementary methods (molecular dynamics simulations, single-molecule in vitro assays and cellular organelle redistribution assays) to investigate the role of the neck linker and cover strand docking in kinesin transport and force generation. They carry out an in-depth computational/structural investigation of the residues in the neck linker, N-term, and catalytic core that come together to dock the cover-neck bundle (CNB) during the kinesin force-generating step. The Lang and Hwang labs have looked into this previously, both through MD simulations and optical trapping experiments, but this work goes beyond that published work. The single-molecule trapping and TIRF data show clearly that mutations in the N-latch and CNB a) decrease force generation capabilities due to premature detachment of the motor, b) increase motor speed and processivity, and c) increase the motor landing rate. The authors also show that the mutant motors are able to redistribute small peroxisomes, whereas they are not able to redistribute the large Golgi. This cellular data is interpreted to be in agreement with the single-molecule results, nicely wrapping up the study. In summary, this manuscript is very interesting, well presented and overall the quality of the data is high. The study is in principle suitable for publication in *eLife*, provided the authors can address the following concerns.

Essential revisions:

1) The classification of peroxisomes and Golgi as low-load and high-load cargos does not appear to be fully justified. It is based on two earlier in vivo trapping studies showing that 2-15 pN force is needed to stall peroxisomes whereas much larger forces are needed for Golgi. While this can be true, there may be many other reasons why mutant proteins can transport peroxisomes but not Golgi to its destination. It is not known how many motors are bound per cargo, or how much force is being produced by these motors per cargo. Therefore, some of the differences they observed might be due to the differences in expression levels, or effectiveness in motor recruitment to a cargo. The sizes of these cargos are different from each other, which might result in different steric effects and drag forces in cytoplasm. In addition, there can be other kinesins engaged in transporting the cargo in the same direction and dynein that transport these cargos towards the opposite direction. These uncertainties should be kept in mind when drawing conclusions, making sure that results are presented separately from interpretations. Editorial revisions are expected to be sufficient to address this point.

2) In this context: Overexpression of dynein DN is done for kinesin-18, but not for the mutants for the Golgi transport. The authors should overexpress dynein DN and see if it helps kinesin-1 mutants to disperse the Golgi, as is the case for kinesin-18, which would support their interpretation. This experiment is doable with the available reagents.

3) Subsection “CNB and N-latch mutants display enhanced unloaded motility properties”: Gaps in kymographs may not be the rapid reattachment of the motor, and may be caused by fluorescence blinking. Although the authors mention that another protein with the same fluorophore does not have these gaps in kymographs, these probes may behave differently when bound/fused to different proteins.

Was there any evidence of fast reattachment in the optical trapping assays? The argument for reattachment solely due to the "gaps" in the unloaded runs is weak and an increased landing rate does not necessarily imply fast reattachment during stepping. In the kinesin-2 optical trapping studies that the authors reference, there were clear reattachment events that may be predicted to be seen for the present mutants.

To clarify the nature of the gaps, the authors could measure the motor run length in force-feedback-controlled optical trap assays under low load and determine whether the motors release quickly under the weak pull of the trap. Given that the Lang lab is fully equipped to do these experiments, the reviewers expect that the authors can test the motility of at least one mutant under one low backward load condition within a reasonable revision time.

4) The authors argue that reduced NL docking in the mutants leads to smaller unbinding forces due to premature detachment. This should however in principle also impede NL docking under zero load. So, in a sense the longer unloaded run lengths and the smaller detachment forces are in conflict. This issue should be addressed either experimentally or by changes to the text. A related point is the observation of pauses and Gaussian rather than exponential run length distributions for the long run length mutants. The authors should clarify how these observations can be explained.

[Editors' note: further revisions were requested prior to acceptance, as described below.]

Thank you for resubmitting your work entitled "Neck linker docking is critical for Kinesin-1 force generation in cells but at a cost to motor speed and processivity" for further consideration at *eLife*. The manuscript has been improved but there are some remaining issues that need to be addressed before acceptance, as outlined below:

The reviewers concluded that 'essential revision point 3' (gaps in kymographs) has not been convincingly addressed: In their view, the new experimental data strengthen the argument that the motors keep walking and that blinking in their fluorescence signal leads to gaps. Small molecules with low Reynolds number do not have momentum. Therefore, they cannot "jump" in the forward direction after releasing from a microtubule, simply because they were walking in that direction. New Figure 4—figure supplement 1 clearly shows that net displacement in the gaps is in the direction of motion and very close to (but slightly lower than) the expected displacement from the constant velocity of the motor before the gap. The evidence is not strong enough to make the conclusion of 'rebinding'. The authors can mention that their results are consistent with (or suggest) motor hopping on a microtubule, but they should also mention that they cannot rule out the possibility of fluorophore blinking leading to gaps in kymograph. This is not one of the main conclusions of this study, so this point can be addressed by softening their argument.

In the light of this remaining criticism, we ask the authors to either tone down their claims, discuss alternative interpretations or even remove (part of the) blinking data/analysis that is not considered convincing by the reviewers.

The reviewers were also not quite satisfied with the response to 'essential revision point 4' but acknowledge that this concerns a complicated topic and is not quite at the center of the study. The authors may however also here consider rewording some of their statements.

---

## [Author Response]

Essential revisions:

*1) The classification of peroxisomes and Golgi as low-load and high-load cargos does not appear to be fully justified. It is based on two earlier* in vivo *trapping studies showing that 2-15 pN force is needed to stall peroxisomes whereas much larger forces are needed for Golgi. While this can be true, there may be many other reasons why mutant proteins can transport peroxisomes but not Golgi to its destination. It is not known how many motors are bound per cargo, or how much force is being produced by these motors per cargo. Therefore, some of the differences they observed might be due to the differences in expression levels, or effectiveness in motor recruitment to a cargo. The sizes of these cargos are different from each other, which might result in different steric effects and drag forces in cytoplasm. In addition, there can be other kinesins engaged in transporting the cargo in the same direction and dynein that transport these cargos towards the opposite direction. These uncertainties should be kept in mind when drawing conclusions, making sure that results are presented separately from interpretations. Editorial revisions are expected to be sufficient to address this point.*

We appreciate the reviewers’ concerns with the cellular assays and have altered the text to reflect these alternative possibilities (Discussion section).

2) In this context: Overexpression of dynein DN is done for kinesin-18, but not for the mutants for the Golgi transport. The authors should overexpress dynein DN and see if it helps kinesin-1 mutants to disperse the Golgi, as is the case for kinesin-18, which would support their interpretation. This experiment is doable with the available reagents.

We have carried out new experiments as suggested by the reviewers (new Figure 7—figure supplement 3). We used overexpression of the dynein DN construct to decrease the load imposed on the motors in the Golgi dispersion assay and found that each of the kinesin-1 mutants (CNB, Latch, and CNB+Latch) is now able to drive Golgi dispersion. Thus, disruption of cytoplasmic dynein activity facilitates transport of the high-load Golgi cargo not only for KIF18A as we showed previously, but also for the CNB, Latch, and CNB+Latch mutant motors.

3) Subsection “CNB and N-latch mutants display enhanced unloaded motility properties”: Gaps in kymographs may not be the rapid reattachment of the motor and may be caused by fluorescence blinking. Although the authors mention that another protein with the same fluorophore does not have these gaps in kymographs, these probes may behave differently when bound/fused to different proteins.Was there any evidence of fast reattachment in the optical trapping assays? The argument for reattachment solely due to the "gaps" in the unloaded runs is weak and an increased landing rate does not necessarily imply fast reattachment during stepping. In the kinesin-2 optical trapping studies that the authors reference, there were clear reattachment events that may be predicted to be seen for the present mutants.

We did not observe evidence of fast reattachment in the optical trapping assays. However, similar to our recent work with the kinesin-14 motor HSET (Reinemann et al., 2018), the low force of the motor and the low stiffness of the trap make any fast reattachment events difficult to record.

To clarify the nature of the gaps, the authors could measure the motor run length in force-feedback-controlled optical trap assays under low load and determine whether the motors release quickly under the weak pull of the trap. Given that the Lang lab is fully equipped to do these experiments, the reviewers expect that the authors can test the motility of at least one mutant under one low backward load condition within a reasonable revision time.

In our email correspondence with the Editor, we relayed the following: “The student who performed the optical trap assays (Nikki Reinemann) has graduated and is in the process of establishing her own lab at the University of Mississippi. There is no one currently in the Lang lab who is familiar with optical trapping assays of kinesin motors and thus able to pull off such a challenging experiment within the revision time. Since the main concern of the reviewers is whether the gaps in the kymographs truly represent fast reattachment rather than fluorescence blinking, we propose the following alternative experiment: we will repeat the single molecule assays for the CNB+Latch mutant (where we see largest number of gaps) tagged with two other fluorophores. First, mRuby (a naturally blinking fluoroscent protein) and second, Janelia Flour 549 dye labeling of a HaloTag. If we see the same gaps in the kymographs regardless of fluorophore blinking, then this will support our hypothesis that the gaps are due to rapid reattachment of the motor rather than fluorescence blinking.”

Following discussion between the Reviewing Editor and the reviewers, the editors agreed and encouraged us to consider the following comment from one of the reviewers: “If the "gaps" in the kymographs are truly detachment and fast reattachment events, then the mean displacement during these events should be zero. In contrast, if they are due to blinking, then the motors should be continuing at the same velocity. By measuring the displacements during the gaps, a finding of near zero or, at a minimum, a displacement significantly less than that expected from steady velocity would be strong evidence. If displacements during dark periods are what is expected from steady velocity, that rules out the "hopping" argument."

We thank the reviewers for this comment. We have carried out new single-molecule imaging experiments for the CNB+Latch mutant motor tagged with HaloTag or mRuby (new Figure 4—figure supplement 1). We find that the kinesin-1 CNB+Latch mutant shows “gaps” in the kymographs regardless of fluorescent tag. When the mutant motor is labeled with the HaloTag JF649 dye (which rarely blinks), the displacement is less than what is expected from steady velocity. In contrast, when the mutant motor is labeled with the mRuby tag (which frequently blinks), some of the gaps display a displacement consistent with steady velocity (due to blinking) and some of the gaps display a displacement less than what is expected from steady velocity (Figure 4—figure supplement 1). We believe this data strongly support our interpretation that the gaps in the kymograph indicate fast reattachment of the mutant motors.

4) The authors argue that reduced NL docking in the mutants leads to smaller unbinding forces due to premature detachment. This should however in principle also impede NL docking under zero load. So, in a sense the longer unloaded run lengths and the smaller detachment forces are in conflict. This issue should be addressed either experimentally or by changes to the text.

It is not clear to us why the reviewer(s) believes that the longer run lengths under no load and the smaller detachment forces under load are in conflict. Explaining why the results are the way they are depends on the model for the motility cycle, coordination among motor heads, and very complex non-equilibrium probabilities leading to rupture in one case and dissociation probability in the other. Perhaps the reviewer has the view that weak force generation equates to weak microtubule binding. Under load, the weakened NL docking may indeed create a weak microtubule binding state that leads to rapid detachment from the microtubule track. However, in the unloaded scenario, weak associations with the microtubule may be sufficient to keep the motor attached. In addition, our MD simulations show that NL docking has allosteric effects on microtubule and nucleotide binding that facilitate motility in the unloaded state.

A related point is the observation of pauses and Gaussian rather than exponential run length distributions for the long run length mutants. The authors should clarify how these observations can be explained.

With respect to the pauses, please see our response above to essential revision #3.

With respect to the run length distributions of the Latch and CNB+Latch mutant motors, have reanalyzed these data fitting to a γ distribution. We believe that the non-exponential distributions of these data indicate that two or more kinetic steps influence motor detachment from the microtubule track during an unloaded run. The identity of these kinetic steps and their mechanistic basis are presently unclear. We note that similar distributions have been reported for other highly processive motors such as kinesin-3 motors (e.g. Soppina et al., 2014, Lessard et al., 2019).

[Editors' note: further revisions were requested prior to acceptance, as described below.]

The reviewers concluded that 'essential revision point 3' (gaps in kymographs) has not been convincingly addressed: In their view, the new experimental data strengthen the argument that the motors keep walking and that blinking in their fluorescence signal leads to gaps. Small molecules with low Reynolds number do not have momentum. Therefore, they cannot "jump" in the forward direction after releasing from a microtubule, simply because they were walking in that direction. New Figure 4—figure supplement 1 clearly shows that net displacement in the gaps is in the direction of motion and very close to (but slightly lower than) the expected displacement from the constant velocity of the motor before the gap. The evidence is not strong enough to make the conclusion of 'rebinding'. The authors can mention that their results are consistent with (or suggest) motor hopping on a microtubule, but they should also mention that they cannot rule out the possibility of fluorophore blinking leading to gaps in kymograph. This is not one of the main conclusions of this study, so this point can be addressed by softening their argument.In the light of this remaining criticism, we ask the authors to either tone down their claims, discuss alternative interpretations or even remove (part of the) blinking data/analysis that is not considered convincing by the reviewers.The reviewers were also not quite satisfied with the response to 'essential revision point 4' but acknowledge that this concerns a complicated topic and is not quite at the center of the study. The authors may however also here consider rewording some of their statements.

In the second revised version, we have addressed the reviewers’ remaining concerns with essential revision #3 by text revisions that include alternative explanations for the observed behavior (gaps in kymographs). With respect to the reviewers’ remaining concerns with essential revision #4, the previously revised version contained altered text to address these concerns and we have now added two additional sentences to the Discussion section.

We thank you for the opportunity to revise our manuscript which has been significantly strengthened by the revisions. We hope that these additional text changes will be satisfactory to the reviewers and the editors.